# Efficient Low Rank Attention for Long-Context Inference in Large Language Models

**Tenghui Li**
Guangdong University of Technology
RIKEN AIP
tenghui.lee@foxmail.com

**Guoxu Zhou** *
Guangdong University of Technology
Key Laboratory of Intelligent Detection and the
Internet of Things in Manufacturing,
Ministry of Education, Guangzhou, CHINA
gx.zhou@gdut.edu.cn

**Xuyang Zhao**
RIKEN iTHEMS
RIKEN IMS
Chiba University
xuyang.zhao@riken.jp

**Yuning Qiu**
RIKEN AIP
yuning.qiu@riken.jp

**Qibin Zhao**
RIKEN AIP
qibin.zhao@riken.jp

## Abstract

As the length of input text increases, the key-value (KV) cache in LLMs imposes prohibitive GPU memory costs and limits long-context inference on resource constrained devices. Existing approaches, such as KV quantization and pruning, reduce memory usage but suffer from numerical precision loss or suboptimal retention of key-value pairs. In this work, Low Rank Query and Key attention (LRQK) is introduced, a two-stage framework that jointly decomposes full-precision query and key matrices into compact rank-$r$ factors during the prefill stage, and then employs these low-dimensional projections to compute proxy attention scores in $\mathcal{O}(lr)$ time at each decode step. By selecting only the top-$k$ tokens and a small fixed set of recent tokens, LRQK employs a mixed GPU-CPU cache with a hit-and-miss mechanism where only missing full-precision KV pairs are transferred, thereby preserving exact attention outputs while reducing CPU-GPU data movement. Extensive experiments on the RULER and LongBench benchmarks with LLaMA-3-8B and Qwen2.5-7B demonstrate that LRQK matches or surpasses leading sparse-attention methods in long context settings, while delivering significant memory savings with minimal accuracy loss. Our code is available at https://github.com/tenghuilee/LRQK.

## 1 Introduction

Large language models (LLMs) have shown their remarkable capabilities across diverse tasks. Recent LLMs have extended their capabilities to support long-context processing, enabling them to leverage extended sequences for improved performance in applications such as document level understanding and long-form text generation [1–5]. Despite these advances, efficient processing of long-contexts presents substantial challenges. The primary constraint stems from computational resource limitations, particularly regarding memory consumption. The KV cache stores historical key-value pairs to prevent redundant computations. However, its size grows linearly with sequence length and quickly imposes prohibitive memory requirements when handling long inputs. Consequently, as context windows expand, the KV cache correspondingly increases in size, eventually becoming the system bottleneck.

---

*corresponding author

39th Conference on Neural Information Processing Systems (NeurIPS 2025).

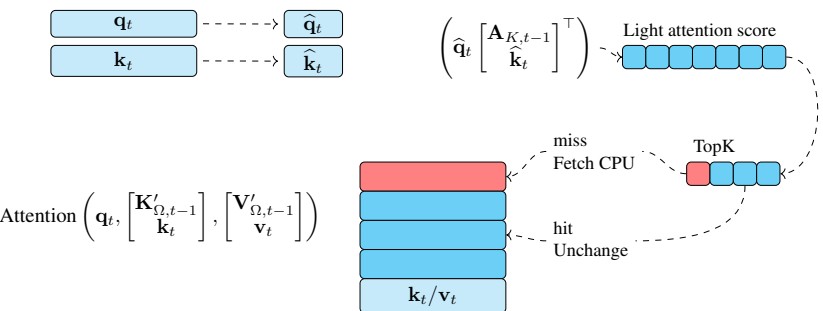

Figure 1: Brief overview of the proposed Low Rank Query and Key attention (LRQK) method. Subscript $\Omega$ denotes the selected tokens, $t$ denotes the current token. $\mathbf{q}_t, \mathbf{k}_t$ are the original query and key, $\widehat{\mathbf{q}}_t, \widehat{\mathbf{k}}_t$ are the approximated query and key. $\mathbf{A}_{K,t}$ is the low rank key matrix. $\mathbf{K}'_{\Omega,t-1}, \mathbf{V}'_{\Omega,t-1}$ are GPU cache $\mathbf{K}_{\Omega,t-1}, \mathbf{V}_{\Omega,t-1}$ merged with fetched CPU keys and values.

Existing approaches addressing the KV cache memory bottleneck can be systematically categorized into three fundamental methodologies: (1) quantization techniques [6–9], which reduce numerical precision while preserving semantic integrity; (2) pruning mechanisms [10–12], which selectively retain critical tokens while eliminating redundant ones; and (3) offloading strategies [13–15], which redistribute memory across heterogeneous storage hierarchies. Additionally, hybrid approaches [16] combine multiple techniques to maximize efficiency.

While these methods have made significant strides in addressing the KV cache memory bottleneck, they each come with inherent limitations. Quantization methods will sacrifice numerical precision, and the evincing techniques risk eliminating potentially crucial key-value pairs, particularly problematic when tokens deemed unimportant in current contexts later become critical for subsequent reasoning. Pruning methods may inadvertently discard important key-value pairs, leading to sub-optimal performance when previously unimportant tokens become critical in subsequent reasoning. Offloading strategies introduce substantial latency overhead due to frequent PCIe data transfers, severely impacting real time processing capabilities.

Consequently, an ideal solution should balance the trade-offs between memory efficiency, numerical precision, and computational latency. A key observation in decoder-only transformer LLMs is the inherent sparsity of attention activation patterns during inference. It had been shown that only a small subset of tokens contributes significant attention weights at each generation step [11, 10, 16, 12]. This sparsity also shows that the possibility of offloading of inactive key-value pairs to cheaper CPU memory while retaining active pairs in GPU memory. However, a fundamental technical challenge emerges when implementing this strategy. Full attention computation requires complete access to key tokens, which are not efficiently accessible after offloading.

To mitigate these challenges, a hybrid attention mechanism is proposed to exploit the intrinsic sparsity of transformer attention patterns while preserving exact computation of attention outputs. At its core lies the observation that full query-key interaction matrix in decoder-only LLMs admits a close low rank approximation. Therefore, the query matrix and key matrix are jointly factorized into compact bases. In the decoding phase, a "proxy" attention score is computed via these factorized query and key matrices, and identifies the top-$k$ relevant tokens in the GPU cache. Thus, the ground truth attention score can be carried out on the retrieved subset of $\mathbf{k}$ and $\mathbf{v}$, guaranteeing numerical fidelity. An overview of the proposed method is shown in Figure 1 and the time comparison is represented in Figure 2. The main contributions of this paper can be summarized as:

- **Joint Low Rank Approximation**: Rather than performing computationally expensive SVD operations on pre-RoPE keys as implemented in ShadowKV [16], the proposed method jointly optimizes low rank approximations of both query and key matrices, and reduces computational complexity while maintaining representation accuracy.

- **Precision-Preserving Attention Computation**: The low rank approximated keys and values serve independently as computational proxies for lightweight attention score estimation. Critically, the subsequent attention layer operations utilize the original query, key, and

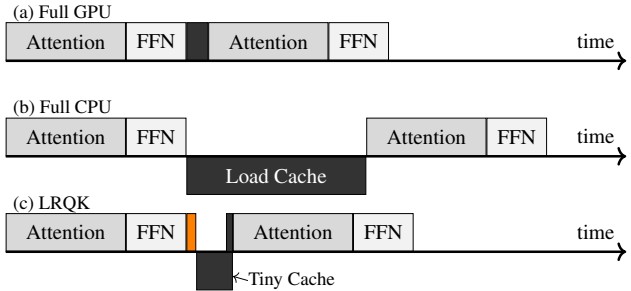

Figure 2: Time comparison with full GPU, full CPU and the proposed LRQK methods. The orange block is the selection operation of KV pairs. The black blocks are cache loading operations. The blocks above line mean GPU operations and the blocks below are CPU operations.

value vectors without any approximation or reconstruction, thereby preserving mathematical fidelity and model performance.

- **Mixed Cache Management**: A sophisticated hybrid GPU-CPU storage system featuring active token retention mechanisms and a hit/miss buffer architecture that minimizes cross device data transfer is implemented. Additionally, a specialized buffer maintains recently accessed keys and values, which empirical analysis confirms consistently receive high attention scores, further optimizing cache hit rates under operational conditions.

The paper is organized as follows. In Section 3, two key insights are presented: the low rank nature of the key matrix and the high attention scores between a current token and its neighbors. Section 4 introduces the Low Rank Query and Key (LRQK) method, detailing both prefill and decode stages. In Section 5, the performance of the LRQK method is assessed across various models and datasets.

## 2 Related Works

### 2.1 Token Eviction

Token eviction methods seek to bound the memory footprint of the KV cache by discarding less informative key-value pairs while preserving generation quality [11, 17, 10, 12, 18, 19]. Early work such as StreamingLLM [11], maintains a fixed "attention sink" of initial tokens together with a sliding window of recent tokens, stabilizing decoding for arbitrarily long sequences. Building on this, $H_2O$ [10] formulates eviction as a dynamic submodule optimization problem to retain both recent tokens and "heavy hitters". Rather than relying on global scoring, SnapKV [12] leverages per-head clustering of attention patterns within a small observation window to select and compress the most relevant KV positions in one pass. More recently, Ada-KV [18] derives a theoretical upper bound on post-eviction attention error and proposes a head-wise adaptive budget allocation strategy, yielding consistent improvements. However, these existing eviction strategies universally trade off either long-range context fidelity or computational simplicity by relying on rigid windows, coarse heuristics, or expensive per-head clustering, therefore resulting in non-negligible approximation error or management overhead. Our approach adopts a joint low-rank proxy to select only the top-$k$ full-precision KV pairs and a small hit/miss recency cache to load them, guaranteeing exact attention outputs and less CPU-GPU data movement.

### 2.2 Dynamic Sparse Attention

Dynamic sparse attention mechanisms represent a significant advancement in transformer based models by selectively computing attention scores on a subset of tokens while maintaining complete key-value cache storage, thereby substantially reducing computational requirements. Contemporary approaches implement diverse token selection strategies with varying degrees of efficacy: QUEST [20] employs a page based selection methodology, while Loki [21] leverages low-dimensional attention approximation through principal component analysis (PCA) on key matrices. PALU [22] and LPA [23] also utility the low rank capability in attention. Notably, two recent techniques, InfiniGen [14] and ShadowKV [16] utilize singular value decomposition (SVD) for key-value selection. InfiniGen

pre-fetches essential entries using predefined projections via SVD, whereas ShadowKV offloads the entire $\mathbf{V}$ cache to CPU memory after SVD decomposition of the key matrix ($\mathbf{K}$). While effective, these SVD based approaches incur substantial computational overhead and fundamentally alter the original query/key representations, potentially compromising model fidelity. In contrast, the proposed method combines a joint low rank $\mathbf{Q}, \mathbf{K}$ projection and retrieves the relevant full-precision KV pairs on demand, ensuring exact attention computation with less approximation error.

## 3 Preliminary

### 3.1 Low Rankness of Query and Key

As demonstrated in prior works [16, 14], the key matrix in decoder-only transformers exhibits a pronounced low rank structure. Motivated by this observation, the full-precision query and key matrices are jointly factorized into rank-$r$ components, enforcing

$$\text{rank}(\mathbf{Q}\mathbf{K}^\top) \leq \min\left(\text{rank}(\mathbf{Q}), \text{rank}(\mathbf{K}^\top)\right) = \min\left(\text{rank}(\mathbf{Q}), \text{rank}(\mathbf{K})\right). \tag{1}$$

This implies that if $\mathbf{K}$ is effectively of rank $r$, then its interaction with any query also lies close to an $r$-dimensional subspace. Consequently, it is possible to approximate $\mathbf{Q}\mathbf{K}^\top$ by a low rank factorization $\mathbf{A}_Q \mathbf{A}_K^\top$ with a low approximation error. An illustrative example of this phenomenon appears in Figure 3, which plots the average singular value spectrum of the per-head query and key matrix $\mathbf{K}$ on Qwen2.5-7B [2] and LLaMA-3-8B-1M [3, 24] under the Wikitext-2-v1 test set [25]. In both cases, the singular values decay rapidly beyond a small rank, confirming that $\mathbf{K}$ admits a low rank approximation with minimal loss.

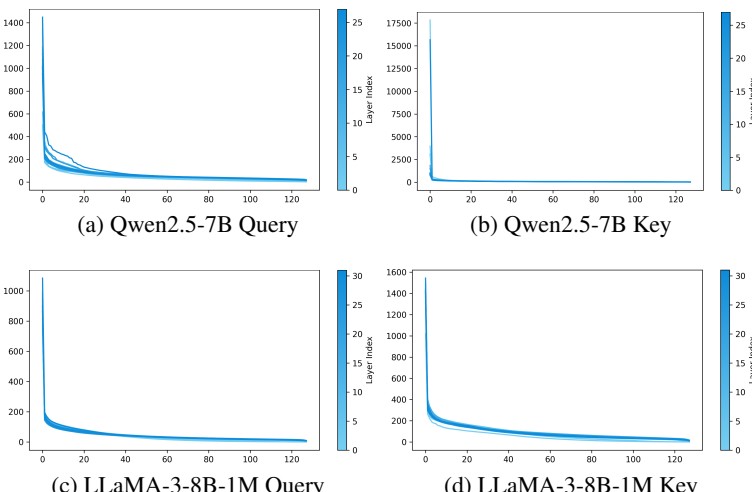

Figure 3: Examples of the mean of singular values of the query and key matrix over different layers on Qwen2.5-7B and LLaMA-3-8B-1M models. The singular values are summed over batches and attention heads.

### 3.2 Attention Scores Near Current Token

To quantify the locality of self-attention during decoding, let $\mathbf{q}_t$ be the query at step $t$ and $\mathbf{k}_i$ the key at position $i$, so that the attention weights are given by

$$\text{softmax}\left(\mathbf{q}_t \begin{bmatrix} \cdots & \mathbf{k}_{t-3} & \mathbf{k}_{t-2} & \mathbf{k}_{t-1} & \mathbf{k}_t \end{bmatrix}\right) = \begin{bmatrix} \cdots & a_{t-3} & a_{t-2} & a_{t-1} & a_t \end{bmatrix}.$$

The average per-head scores are evaluated on the Wikitext-2-v1 test set using both Qwen2.5-7B and LLaMA-3-8B-1M. Results are shown in Figure 4. In every case, the curves exhibit pronounced higher attention scores in the current token and its near neighbors, confirming a strong recency bias. This empirical finding is also observed by StreamingLLM [11], which motivates our inclusion of a compact recency buffer in the GPU cache that persistently holds the most recent tokens, thereby maximizing cache hit rates with minimal overhead.

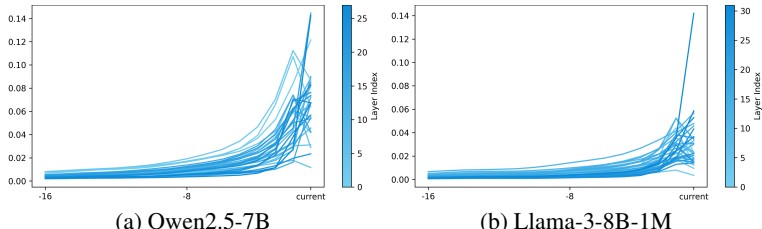

(a) Qwen2.5-7B          (b) Llama-3-8B-1M

Figure 4: Examples of the attention scores of the neighbors of current token. The window size is 16. The $x$-label "current" is the index of the current token, $-8$ means the tokens at $t - 8$, and so on. The attention scores are averaged over batches and attention heads.

# 4 Proposed Method

The inference process of LLM can be divided into two distinct phases: prefill and decode. When a prompt is provided by the user, the long prompt is first prefilled once by the LLM, and then the token is decoded autoregressively. Specific cache mechanisms are required for these two phases, the LRQK will be introduced for prefill in Section 4.2 and decode in Section 4.3, respectively.

The preliminaries and formulation used in the proposed method are clarified below. In practical implementations, transformer models typically process input tensors with a shape of (batch size, num heads, sequence length, hidden size). However, when methods such as GQA [26] are employed, the number of attention heads used for the query tensor is usually different from those used for the key and value tensors. Therefore, repeated key/value (repeat KV) mechanism is utilized, ensuring that the number of heads for the query and the key/value tensors are aligned. In addition, to simplify notation, 'batch size' and 'num heads' dimensions are also omitted, as computations across different batches and heads are independent. Thus, the query, key, and value matrices as $\mathbf{Q}, \mathbf{K}, \mathbf{V} \in \mathbb{R}^{l \times d}$, where $l$ denotes the sequence length and $d$ the hidden size of each head. Moreover, since the prefill and decode process is identical across all transformer layers, the layer indices are omitted for clarity.

## 4.1 Mixed Caching

To address memory constraints and optimize runtime efficiency, a mixed caching strategy is adopted that leverages both a fast yet limited GPU KV cache and a larger but slower CPU cache. To further reduce the data transfer overhead between GPU and CPU, a hit-and-miss cache mechanism is introduced. In this scheme, when the required tokens are already present in the GPU cache (i.e., a cache hit), direct access is provided without incurring additional transfer cost. When the required cache is missing in the GPU, only the missing tokens are fetched from the CPU cache, therefore avoiding redundant data movement and improving overall system efficiency.

Based on the observation that recent tokens are more likely to receive high attention scores (as shown in Figure 4), a dedicated buffer is incorporated to store the most recent tokens. The GPU KV cache is structured as $[\quad$ active tokens $\quad$ lite tokens $\quad]$, where the 'active tokens' are the top-$k$ tokens corresponding to the set $\Omega_k$ with the approximated highest attention scores selected by the low rank approximations of query and key matrices, and the 'lite tokens' corresponding to the set $\Omega_l$ capture a fixed number of neighboring recent tokens. The total window size of the GPU KV cache is thus the sum of the active and lite tokens, allowing for a balance between relevance and recency in cache composition.

## 4.2 Prefill

At this stage, the input consists of a long prompt represented as a matrix $\mathbf{X} \in \mathbb{R}^{l \times d}$, where $l$ denotes the sequence length and $d$ the hidden dimension. By applying linear projections to $\mathbf{X}$, the query, key, and value matrices: $\mathbf{Q}, \mathbf{K}, \mathbf{V} \in \mathbb{R}^{l \times d}$ will be obtained.

According to (1), the attention score matrix is usually low rank, thus can be approximated using a jointly low rank decomposition:

$$\mathbf{Q}\mathbf{K}^{\top} \approx \mathbf{A}_Q \mathbf{A}_K^{\top}, \tag{2}$$

where $\mathbf{A}_Q, \mathbf{A}_K \in \mathbb{R}^{l \times r}$ and $r \ll d$ denotes matrix rank. In addition to the joint low rank approximation, query and key matrices are approximated for memory efficiency:

$$\mathbf{Q} \approx \mathbf{A}_Q \mathbf{B}_Q, \quad \mathbf{K} \approx \mathbf{A}_K \mathbf{B}_K, \tag{3}$$

where $\mathbf{B}_Q, \mathbf{B}_K \in \mathbb{R}^{r \times d}$. Note that the approximation for attention score $\mathbf{Q}\mathbf{K}^\top$ and $\mathbf{Q}, \mathbf{K}$ shares the same column/row space spanned by $\mathbf{A}_Q$ and $\mathbf{A}_K$. For solving (2) and (3), it can be formulated as

$$\underset{\mathbf{A}_Q, \mathbf{B}_Q, \mathbf{A}_K, \mathbf{B}_K}{\arg \min} \frac{1}{2} \left\| \mathbf{Q}\mathbf{K}^\top - \mathbf{A}_Q \mathbf{A}_K^\top \right\|_{\mathrm{F}}^2, \text{s.t. } \mathbf{Q} = \mathbf{A}_Q \mathbf{B}_Q, \mathbf{K} = \mathbf{A}_K \mathbf{B}_K. \tag{4}$$

To solve the constraint optimization problem (4), it is relaxed by deriving a Lagrangian:

$$\mathcal{L}_{pre} = \frac{1}{2} \left\| \mathbf{Q}\mathbf{K}^\top - \mathbf{A}_Q \mathbf{A}_K^\top \right\|_{\mathrm{F}}^2 + \frac{\lambda_{pQ}}{2} \left\| \mathbf{Q} - \mathbf{A}_Q \mathbf{B}_Q \right\|_{\mathrm{F}}^2 + \frac{\lambda_{pK}}{2} \left\| \mathbf{K} - \mathbf{A}_K \mathbf{B}_K \right\|_{\mathrm{F}}^2, \tag{5}$$

where $\lambda_{pQ}$ and $\lambda_{pK}$ are two scaling factors indicating the importance of the low rank approximation of the query and key matrices, respectively.

By solving the Lagrangian (5), with setting the partial derivative to be zero, the solution of the query and key matrices can be obtained (see Appendix A.1 for details):

$$
\begin{aligned}
\mathbf{A}_Q &= \mathbf{Q} \left( \mathbf{K}^\top \mathbf{A}_K + \lambda_{pQ} \mathbf{B}_Q^\top \right) \left( \mathbf{A}_K^\top \mathbf{A}_K + \lambda_{pQ} \mathbf{B}_Q \mathbf{B}_Q^\top \right)^{-1} \\
\mathbf{A}_K &= \mathbf{K} \left( \mathbf{Q}^\top \mathbf{A}_Q + \lambda_{pK} \mathbf{B}_K^\top \right) \left( \mathbf{A}_Q^\top \mathbf{A}_Q + \lambda_{pK} \mathbf{B}_K \mathbf{B}_K^\top \right)^{-1} \\
\mathbf{B}_Q &= (\mathbf{A}_Q^\top \mathbf{A}_Q)^{-1} \mathbf{A}_Q^\top \mathbf{Q} \\
\mathbf{B}_K &= (\mathbf{A}_K^\top \mathbf{A}_K)^{-1} \mathbf{A}_K^\top \mathbf{K}.
\end{aligned}
\tag{6}
$$

From (6), it is notice that the operation of matrix inverse is required. The matrix inverse operation is expensive $O(r^3)$. However, since the rank $r$ is a small number, and $\mathbf{A}_K^\top \mathbf{A}_K, \mathbf{A}_Q^\top \mathbf{A}_Q$, $\mathbf{B}_K \mathbf{B}_K^\top, \mathbf{B}_Q \mathbf{B}_Q^\top$ are all with shape $(r, r)$, which are small matrices. Therefore, the computation costs of matrix inverse are relatively low. In addition, when solving $\mathbf{A}_Q$ and $\mathbf{A}_K$, the memory cost can further be reduced by changing the multiplication order. For example, by changing the order of operations from $(\mathbf{Q}\mathbf{K}^\top)\mathbf{A}_K$ to $\mathbf{Q}(\mathbf{K}^\top \mathbf{A}_K)$, the computational complexity is reduced from $\mathcal{O}(l^2 d)$ to $\mathcal{O}(rld)$.

The iterative procedure in Algorithm 1 can be interpreted as a Block Coordinate Descent (BCD) method, where each step optimizes the Lagrangian with respect to a subset of variables while holding the others fixed.

---

**Algorithm 1** Prefill of LRQK, alternating updates for $\mathbf{A}_Q, \mathbf{A}_K, \mathbf{B}_Q, \mathbf{B}_K$

---

**Require:** $\mathbf{Q}, \mathbf{K}$, rank $r$, $\lambda_{pQ}, \lambda_{pK}$
1: Initialize $\mathbf{A}_Q, \mathbf{A}_K \sim \mathcal{N}(0, 1)$
2: **while** $i <$ max iteration or not converge **do**
3:      Update $\mathbf{B}_Q \leftarrow (\mathbf{A}_Q^\top \mathbf{A}_Q)^{-1} \mathbf{A}_Q^\top \mathbf{Q}$
4:      Update $\mathbf{B}_K \leftarrow (\mathbf{A}_K^\top \mathbf{A}_K)^{-1} \mathbf{A}_K^\top \mathbf{K}$
5:      Update $\mathbf{A}_K \leftarrow \mathbf{K} \left( \mathbf{Q}^\top \mathbf{A}_Q + \lambda_{pK} \mathbf{B}_K^\top \right) \left( \mathbf{A}_Q^\top \mathbf{A}_Q + \lambda_{pK} \mathbf{B}_K \mathbf{B}_K^\top \right)^{-1}$
6:      Update $\mathbf{A}_Q \leftarrow \mathbf{Q} \left( \mathbf{K}^\top \mathbf{A}_K + \lambda_{pQ} \mathbf{B}_Q^\top \right) \left( \mathbf{A}_K^\top \mathbf{A}_K + \lambda_{pQ} \mathbf{B}_Q \mathbf{B}_Q^\top \right)^{-1}$
7: **end while**
8: **return** $\mathbf{A}_Q, \mathbf{A}_K, \mathbf{B}_Q, \mathbf{B}_K$

---

### 4.3 Decode

Following the prefilling phase, the language model generates the remaining tokens sequentially, adhering to an autoregressive decoding process where each token is predicted based on the previously generated context. Suppose the current stage is $t$, and the input is a row vector $\mathbf{x}_t \in \mathbb{R}^{1 \times d}$ and its projections are $\mathbf{q}_t \in \mathbb{R}^{1 \times d}$ and $\mathbf{k}_t \in \mathbb{R}^{1 \times d}$. In the decode stage, the goal is to compress the $\mathbf{q}_t$ and $\mathbf{k}_t$

to much smaller ones $\widehat{\mathbf{q}}_t \in \mathbb{R}^{1 \times r}$ and $\widehat{\mathbf{k}}_t \in \mathbb{R}^{1 \times r}$, therefore the memory footprint of KV cache can be greatly mitigated.

From the previous stage, the low rank approximations of the query and key matrices are obtained in the form of left and right factors: $\mathbf{A}_Q, \mathbf{A}_K \in \mathbb{R}^{l \times r}$, and $\mathbf{B}_Q, \mathbf{B}_K \in \mathbb{R}^{r \times d}$. To maintain consistency in the modeling setup for new tokens, the following optimization problem is formulated for estimating the approximated query and key vectors, $\widehat{\mathbf{q}}_t$ and $\widetilde{\mathbf{k}}_t$, respectively,

$$
\underset{\widehat{\mathbf{q}}_t, \widehat{\mathbf{k}}_t}{\arg\min} \frac{1}{2} \|\widehat{\mathbf{q}}_t \mathbf{B}_{Q,t-1} - \mathbf{q}_t\|_{\mathrm{F}}^2 + \frac{1}{2} \left\|\widehat{\mathbf{k}}_t \mathbf{B}_{K,t-1} - \mathbf{k}_t\right\|_{\mathrm{F}}^2,
$$
$$
\text{s.t. } \widehat{\mathbf{q}}_t \widehat{\mathbf{k}}_t^\top = \mathbf{q}_t \mathbf{k}_t^\top, \widehat{\mathbf{q}}_t \mathbf{A}_{K,\Omega,t-1}^\top = \mathbf{q}_t \mathbf{K}_{\Omega,t-1}^\top. \tag{7}
$$

The matrix $\mathbf{A}_{K,\Omega,t-1}$ denotes a submatrix of $\mathbf{A}_K$ that includes only the rows indexed by $\Omega$. Similarly, $\mathbf{K}_{\Omega,t-1}$ is the corresponding submatrix of $\mathbf{K}$ formed by selecting the rows indexed by $\Omega$. Here, $\Omega$ represents the index set of KV cache retained in GPU memory at time step $t-1$.

The Lagrangian of (7) can be written as:

$$
\mathcal{L}_{\mathrm{dec}} = \frac{1}{2} \|\widehat{\mathbf{q}}_t \mathbf{B}_{Q,t-1} - \mathbf{q}_t\|_{\mathrm{F}}^2 + \frac{1}{2} \left\|\widehat{\mathbf{k}}_t \mathbf{B}_{K,t-1} - \mathbf{k}_t\right\|_{\mathrm{F}}^2
$$
$$
+ \frac{\lambda_{d1}}{2} \left(\widehat{\mathbf{q}}_t \widehat{\mathbf{k}}_t^\top - \mathbf{q}_t \mathbf{k}_t^\top\right)^2 + \frac{\lambda_{d2}}{2} \left\|\widehat{\mathbf{q}}_t \mathbf{A}_{K,\Omega,t-1}^\top - \mathbf{q}_t \mathbf{K}_{\Omega,t-1}^\top\right\|_{\mathrm{F}}^2, \tag{8}
$$

where $\lambda_{d1}$ and $\lambda_{d2}$ are regularization parameters. For more details, please refer to the Appendix A.2.

By solving the Lagrangian (8), update rules for $\widehat{\mathbf{q}}_t$ and $\widehat{\mathbf{k}}_t$ can be obtained as:

$$
\widehat{\mathbf{q}}_t = \mathbf{m}_{lq,t} \mathbf{M}_{rq,t}^{-1},
$$
$$
\widehat{\mathbf{k}}_t = \left(\mathbf{k}_t \mathbf{B}_{K,t-1}^\top + \lambda_{d1} \mathbf{k}_t \mathbf{q}_t^\top \widehat{\mathbf{q}}_t\right) \left(\mathbf{B}_{K,t-1} \mathbf{B}_{K,t-1}^\top + \lambda_{d1} \widehat{\mathbf{q}}_t^\top \widehat{\mathbf{q}}_t\right)^{-1}
$$
$$
\mathbf{m}_{lq,t} := \left(\mathbf{q}_t \mathbf{B}_{Q,t-1}^\top + \lambda_{d1} \mathbf{q}_t \mathbf{k}_t^\top \widehat{\mathbf{k}}_t + \lambda_{d2} \mathbf{q}_t \mathbf{K}_{\Omega,t-1}^\top \mathbf{A}_{K,\Omega,t-1}\right) \tag{9}
$$
$$
\mathbf{M}_{rq,t} := \left(\mathbf{B}_{Q,t-1} \mathbf{B}_{Q,t-1}^\top + \lambda_{d1} \widehat{\mathbf{k}}_t^\top \widehat{\mathbf{k}}_t + \lambda_{d2} \mathbf{A}_{K,\Omega,t-1}^\top \mathbf{A}_{K,\Omega,t-1}\right).
$$

Similar to prefill stage, since the matrices require taking inverse is of size $(r, r)$, which is small, and the computation cost is relatively small.

For $\mathbf{B}_{Q,t}$ and $\mathbf{B}_{K,t}$, one step gradient descent is applied to op the objective function. The update rules are given by:

$$
\mathbf{B}_{Q,t} \leftarrow \mathbf{B}_{Q,t-1} - \eta_Q^* \nabla \mathbf{B}_{Q,t-1}, \quad \mathbf{B}_{K,t} \leftarrow \mathbf{B}_{K,t-1} - \eta_K^* \nabla \mathbf{B}_{K,t-1}, \tag{10}
$$

where $\eta_Q$ and $\eta_K$ are the learning rates for $\mathbf{B}_{Q,t-1}$ and $\mathbf{B}_{K,t-1}$, respectively. Two learning rates $\eta_Q^*$ and $\eta_K^*$ are computed as:

$$
\eta_Q^* = \frac{(\widehat{\mathbf{q}}_t \mathbf{B}_{Q,t-1} - \mathbf{q}_t)(\widehat{\mathbf{q}}_t \nabla \mathbf{B}_{Q,t-1})^\top}{(\widehat{\mathbf{q}}_t \nabla \mathbf{B}_{Q,t-1})(\widehat{\mathbf{q}}_t \nabla \mathbf{B}_{Q,t-1})^\top}, \tag{11}
$$

$$
\eta_K^* = \frac{\left(\widehat{\mathbf{k}}_t \mathbf{B}_{K,t-1} - \mathbf{k}_t\right)\left(\widehat{\mathbf{k}}_t \nabla \mathbf{B}_{K,t-1}\right)^\top}{\left(\widehat{\mathbf{k}}_t \nabla \mathbf{B}_{K,t-1}\right)\left(\widehat{\mathbf{k}}_t \nabla \mathbf{B}_{K,t-1}\right)^\top}. \tag{12}
$$

Algorithm 2 summarizes the update rules for $\widehat{\mathbf{q}}_t$, $\widehat{\mathbf{k}}_t$, $\mathbf{B}_{Q,t}$, and $\mathbf{B}_{K,t}$, the computation of proxy attention, and the update for selected key and value. The graphical representation of the thorough decoding process of LRQK is shown in Figure 1. To avoid random initialization, the initial guess of $\widehat{\mathbf{k}}_t$ is utilized by setting $\lambda_{d1} = 0$. The algorithm iterates until convergence or reaches the maximum number of iterations. The attention is computed with selected key and value, Attention $(\mathbf{q}_t, \mathbf{K}_{\Omega,t}, \mathbf{V}_{\Omega,t})$.

**Algorithm 2** Decode of LRQK

**Require:** $\mathbf{q}_t, \mathbf{k}_t, \mathbf{v}_t, \mathbf{B}_{Q,t-1}, \mathbf{B}_{K,t-1}, \lambda_{d1}, \lambda_{d2}, \mathbf{A}_{K,\Omega,t-1}, \mathbf{K}_{\Omega,t-1}, \mathbf{V}_{\Omega,t-1}$

1: Initial Guess $\widehat{\mathbf{k}}_t \leftarrow \left(\mathbf{k}_t \mathbf{B}_{K,t-1}^\top\right)\left(\mathbf{B}_{K,t-1}\mathbf{B}_{K,t-1}^\top\right)^{-1}$
2: **while** $i <$ max iteration or not converged **do**
3:     Update $\widehat{\mathbf{q}}_t \leftarrow \mathbf{m}_{lq,t}\mathbf{M}_{rq,t}^{-1}$
4:     Update $\widehat{\mathbf{k}}_t \leftarrow \left(\mathbf{k}_t \mathbf{B}_{K,t-1}^\top + \lambda_{d1}\mathbf{q}_t\mathbf{k}_t^\top\widehat{\mathbf{q}}_t\right)\left(\mathbf{B}_{K,t-1}\mathbf{B}_{K,t-1}^\top + \lambda_{d1}\widehat{\mathbf{q}}_t^\top\widehat{\mathbf{q}}_t\right)^{-1}$
5: **end while**
6: $\mathbf{B}_{Q,t} \leftarrow \mathbf{B}_{Q,t-1} - \eta_Q^*\nabla\mathbf{B}_{Q,t-1}$, with $\eta_Q^*$ in (11)
7: $\mathbf{B}_{K,t} \leftarrow \mathbf{B}_{K,t-1} - \eta_K^*\nabla\mathbf{B}_{K,t-1}$, with $\eta_K^*$ in (12)
8: $\mathbf{A}_{K,t} \leftarrow \left[\mathbf{A}_{K,t-1};\widehat{\mathbf{k}}_t\right]$                                ▷ concatenate
9: top-$k$ index $\Omega_k \leftarrow \text{top}\left(\mathbf{A}_{K,t}\widehat{\mathbf{q}}_t^\top, k\right)$               ▷ compute proxy attention
10: Fetch missing $\{\mathbf{k}_i\}, \{\mathbf{v}_i\}$ from CPU $\mathbf{K}, \mathbf{V}$ according to $\Omega_k$
11: $\mathbf{K}'_{\Omega,t-1}, \mathbf{V}'_{\Omega,t-1} \leftarrow$ merge fetched $\{\mathbf{k}_i\}, \{\mathbf{v}_i\}$ with GPU $\mathbf{K}_{\Omega,t-1}, \mathbf{V}_{\Omega,t-1}$
12: $\mathbf{K}_{\Omega,t} \leftarrow \left[\mathbf{K}'_{\Omega,t-1};\mathbf{k}_t\right], \mathbf{V}_{\Omega,t} \leftarrow \left[\mathbf{V}'_{\Omega,t-1};\mathbf{v}_t\right]$
13: Async $\mathbf{k}_t, \mathbf{v}_t$ to CPU
14: **return** $\mathbf{K}_{\Omega,t}, \mathbf{V}_{\Omega,t}$.

# 5 Experiments

In this section, a series of experiments are conducted to rigorously evaluate the effectiveness and performance of our proposed method. The experiments are performed on NVIDIA A100 GPUs. The number of max iteration and the tolerance introduced in Algorithm 1 and 2 are chosen as 2 and 0.01, respectively. All scaling parameters are set as $\lambda_{pQ} = \lambda_{pK} = \lambda_{d1} = \lambda_{d2} = 1$. All LLM evaluations are empowered by OpenCompass [27]. Additional results can be found in Appendix B, the time cost analysis is presented in Appendix C, and a guideline for hyperparameter tuning is provided in Appendix D.

## 5.1 Accuracy Evaluation

The accuracy of the proposed method is evaluated on the RULER dataset [28], using a long-context setting with a sequence length of 128K tokens. For the low rank approximation, the rank is set to $r = 32$. The number of top-$k$ tokens selected based on attention scores is set to 2048 (1.56% of 128K), while the number of lite tokens (the most recently generated tokens) is set to 64. The backbone language model used in the evaluation is LLaMA-3-8B-1M [24]. This method is compared against four recent dynamic sparse attention baselines: Loki [21], InfiniGen [14], Quest [20], and ShadowKV [16]. Results are shown in Table 1.

Table 1: Comparison of different models and methods on RULER (left) and LongBench (right).

| Methods | S1 | S2 | MK1 | MK2 | MQ | MV | QA-1 | QA-2 | VT | FWE | NQA | MQA | GRep | SAM | PRetr | LCC |
|---|---|---|---|---|---|---|---|---|---|---|---|---|---|---|---|---|
| Llama-3-8B-1M | 100.00 | 100.00 | 98.96 | 98.96 | 98.96 | 95.57 | 75.00 | 48.96 | 78.54 | 71.85 | 18.98 | 41.84 | 34.18 | 35.96 | 81.50 | 56.07 |
| Loki | 18.75 | 1.04 | 2.08 | 0.00 | 1.56 | 0.78 | 4.17 | 13.54 | 26.04 | 25.35 | 2.26 | 10.19 | 28.97 | 7.84 | 40.52 | 31.44 |
| InfiniGen | 100.00 | 98.96 | 84.38 | 53.13 | 63.28 | 54.95 | 65.63 | 48.96 | **81.67** | 50.35 | 14.39 | 31.46 | 27.38 | 21.97 | 74.30 | 38.58 |
| Quest | 100.00 | 100.00 | **98.96** | 77.08 | 97.65 | 93.49 | 60.42 | 50.00 | 77.08 | 65.63 | **20.13** | 36.63 | 27.11 | 35.63 | 79.00 | 53.64 |
| ShadowKV | 100.00 | 100.00 | 97.92 | **98.96** | 96.88 | 95.83 | 72.92 | 52.08 | **81.67** | **72.57** | 17.17 | 39.73 | **31.62** | **35.87** | 80.00 | 63.93 |
| Proposed | 81.00 | 100.00 | 97.00 | 42.00 | **99.25** | **98.00** | 75.00 | **53.00** | 80.20 | 69.67 | 16.52 | **40.48** | 20.40 | 26.35 | **89.00** | **66.13** |

Table 1 presents the comparative performance of various models and methods on both the RULER and LongBench benchmarks. Our proposed method demonstrates competitive results across a wide range of tasks, particularly excelling in several key domains.

On the RULER benchmark, our model achieves perfect accuracy on the S2 (NIAH Single 2) task, matching the performance of larger models like ShadowKV and Quest. Notably, our method outperforms all baselines on QA-1 (QA SQuAD) and QA-2 (QA HotpotQA). In the MQ (NIAH MultiQuery) and MV (NIAH MultiValue) tasks, which evaluate a model's ability to process multiple

queries or values within a single instance, our method even surpasses the original LLaMA-3-8B-1M model, achieving results of 99.25% and 98.00%, respectively.

For the LongBench evaluation, our approach demonstrates notable gains in tasks such as PRetr (Passage Retrieval) and LCC, outperforming all other methods with 89.00% and 66.13%, respectively. These tasks are particularly demanding due to their emphasis on retrieving and reasoning over long-range dependencies, suggesting that our method is well-suited for these kinds of tasks.

While some tasks like MK2 (NIAH MultiKey 2) and NQA (NarrativeQA) show slightly lower performance compared to leading baselines, the overall consistency and high scores across multiple complex tasks underscore the effectiveness of our approach. These results affirm the potential of our method in addressing long-context language understanding challenges.

The proposed method is further evaluated on two backbone models: LLaMA-3-8B-1M [24] and Qwen2.5-7B-Instruct [2], using the same configuration of rank $r = 16$, top-$k = 256$ (6.25% of 4K), and 16 lite tokens on a subset of RULER-4K. As shown in Table 2, our method consists the performance of both models.

Table 2: Results on subset of RULER-4K of two models

| Methods | QA-1 | QA-2 | VT | Methods | QA-1 | QA-2 | VT |
|---|---|---|---|---|---|---|---|
| LLaMA-3-8B-1M | 82.00 | 58.00 | 99.00 | Qwen2.5-7B | 90.00 | 64.00 | 99.20 |
| + LRQK | 84.00 | 57.00 | 98.80 | + LRQK | 91.00 | 65.00 | 96.60 |

## 5.2 Impact of Rank and Top-$k$ Selection

In this subsection, the impact of two key hyperparameters in the proposed method are investigated: the rank $r$ used in the low rank approximation, and the number of top-$k$ active tokens selected based on attention scores.

**Rank Selection.** To isolate the effect of the rank, we fix the number of top-$k$ tokens at 256 and the number of lite tokens at 16. The rank are vary over the set $\{16, 24, 32\}$, allowing us to examine how the capacity of the low rank representations affects model performance.

**Top-$k$ Selection.** To analyze the influence of the number of active tokens, we fix the rank at 8 and the number of lite tokens at 16. The top-$k$ parameter is varied across $\{256, 512, 1024\}$ to evaluate how the size of the active set impacts retrieval effectiveness and downstream task accuracy. All experiments are conducted on a representative subset of the RULER-4K dataset using the LLaMA-3-8B-1M model as the backbone.

Tables 3 and 4 show that increasing the rank $r$ generally improves performance on QA-1 and VT, with $r = 32$ even surpassing the original LLaMA-3-8B-1M on QA-1. In contrast, QA-2 performs best at a lower rank $r = 16$, suggesting some tasks benefit from more compact representations. For top-$k$, larger values consistently lead to better accuracy across all tasks. The configuration with $k = 1024$ achieves the highest scores, indicating that a broader active token window improves context modeling. However, larger top-$k$ tokens will increase both computation and memory costs, necessitating a trade-off between performance and resource usage. Overall, while higher ranks and larger top-$k$ values enhance performance, smaller settings (e.g., $r = 16$, $k = 512$) still offer strong results with reduced resource usage.

Table 3: Accuracy across different rank values with top-$k$ 256 and 16 lite tokens

| Models | QA-1 | QA-2 | VT |
|---|---|---|---|
| Llama-3-8B-1M | 82.00 | 58.00 | 99.00 |
| $r = 8$ | 79.00 | 50.00 | 56.80 |
| $r = 16$ | 80.00 | 60.00 | 98.80 |
| $r = 24$ | 83.00 | 56.00 | 98.80 |
| $r = 32$ | 84.00 | 57.00 | 99.00 |

Table 4: Accuracy across different top-$k$ values with $r = 8$ and 16 lite tokens

| Models | QA-1 | QA-2 | VT |
|---|---|---|---|
| Llama-3-8B-1M | 82.00 | 58.00 | 99.00 |
| top-256 | 79.00 | 50.00 | 56.80 |
| top-512 | 78.00 | 61.00 | 95.60 |
| top-1024 | 83.00 | 63.00 | 100.00 |

### 5.3 Miss Rates

The performance of the proposed strategy for memory management is evaluated by computing the miss rate. The rate is a ratio $c_{miss}/c_{total}$, where $c_{miss}$ denotes the number of KV cache rows that must be transferred from CPU to GPU, and $c_{total}$ is the number of all selected indices. This metric quantifies the efficiency of the token selection mechanism in reducing memory transfers.

Experiments are conducted on the summarization task using the wikitext-2-v1 test set [25]. A grid search is performed over different hyperparameters, including the rank $r \in \{8, 16, 32, 64\}$, the number of active tokens $\in \{128, 256, 512\}$, and the number of lite tokens $\in \{4, 8\}$. The distribution of miss rates is illustrated in Figure 5, which reveals an approximately Gaussian shape with a mean miss rate of around 0.40 and a standard deviation of approximately 0.10. This implies that, on average, the proposed hit-and-miss mechanism reduces the amount of data transferred from CPU to GPU by about 60% in average.

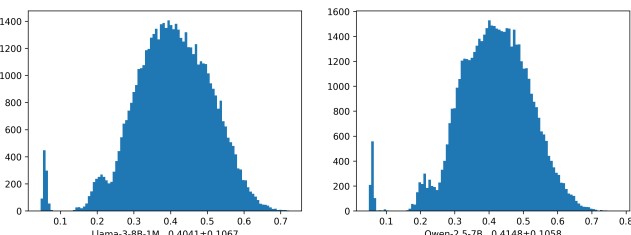

Figure 5: Histogram of miss rates on wikitext-2-v1.

## 6  Conclusion

Low Rank Query and Key (LRQK) attention, a two stage inference algorithm that enables long-context processing through joint low rank decomposition of query and key matrices combined with mixed GPU-CPU cache management is presented. By computing proxy attention scores on compact rank-$r$ factors and selectively fetching only the top-$k$ active tokens and most recent tokens' full-precision key-value pairs, LRQK preserves exact attention outputs while reducing CPU-GPU data transfers.

Extensive evaluations on RULER (up to 128K tokens) or LongBench benchmarks demonstrate that LRQK matches or exceeds state-of-the-art sparse attention methods across diverse long-context tasks. The method achieves substantial memory savings, enabling processing of contexts that would otherwise cause out-of-memory errors, without significant accuracy degradation. Ablation studies confirm the effectiveness of each component: low rank approximation, active token selection, and lite token retention all contribute to the method's robust performance.

**Limitations.**  While LRQK significantly reduces data transfer overhead, further observation reveals that CPU-side indexing operations constitute the primary performance bottleneck rather than PCIe bandwidth limitations, and hoping can be addressed in future work. Additionally, LRQK's hyper-parameters require task-specific tuning. While a practical guidelines and default configurations are provided, optimal settings still demand empirical validation for new domains.

**Societal Impact.**  LRQK's reduced inference costs could enable broader access to long-context language models, benefiting research and education. However, this efficiency may also lower barriers to deploying harmful applications such as large-scale disinformation campaigns. We believe democratized access to advanced AI capabilities provides net benefits when coupled with appropriate safeguards and responsible deployment practices.

## Acknowledgments and Disclosure of Funding

This study was supported in part by Natural Science Foundation of China under Grant 62203124, JSPS KAKENHI Grant Number JP23K28109 and JP24K20849, and Moonshot R&D Grant Number JPMJMS2239. Yuning Qiu was supported by the RIKEN Special Postdoctoral Researcher Program. Tenghui Li was supported by RIKEN's IPA Program.

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

# Appendix

## A Details of Derivatives

### A.1 Prefill Derivatives

The Lagrangian (5) is formulated as:

$$\mathcal{L}_{pre} = \frac{1}{2} \left\| \mathbf{Q}\mathbf{K}^\top - \mathbf{A}_Q\mathbf{A}_K^\top \right\|_{\mathrm{F}}^2 + \frac{\lambda_{pQ}}{2} \left\| \mathbf{Q} - \mathbf{A}_Q\mathbf{B}_Q \right\|_{\mathrm{F}}^2 + \frac{\lambda_{pK}}{2} \left\| \mathbf{K} - \mathbf{A}_K\mathbf{B}_K \right\|_{\mathrm{F}}^2,$$

Compute partial derivative:

$$
\begin{aligned}
\frac{\partial \mathcal{L}_{pre}}{\partial \mathbf{A}_Q} &= -\left( \mathbf{Q}\mathbf{K}^\top - \mathbf{A}_Q\mathbf{A}_K^\top \right) \mathbf{A}_K - \lambda_{pQ} \left( \mathbf{Q} - \mathbf{A}_Q\mathbf{B}_Q \right) \mathbf{B}_Q^\top \\
\frac{\partial \mathcal{L}_{pre}}{\partial \mathbf{A}_K} &= -\left( \mathbf{Q}\mathbf{K}^\top - \mathbf{A}_Q\mathbf{A}_K^\top \right)^\top \mathbf{A}_Q - \lambda_{pK} \left( \mathbf{K} - \mathbf{A}_K\mathbf{B}_K \right) \mathbf{B}_K^\top \\
\frac{\partial \mathcal{L}_{pre}}{\partial \mathbf{B}_Q} &= -\lambda_{pQ}\mathbf{A}_Q^\top \left( \mathbf{Q} - \mathbf{A}_Q\mathbf{B}_Q \right) \\
\frac{\partial \mathcal{L}_{pre}}{\partial \mathbf{B}_K} &= -\lambda_{pQ}\mathbf{A}_K^\top \left( \mathbf{K} - \mathbf{A}_K\mathbf{B}_K \right)
\end{aligned}
\tag{13}
$$

Setting the partial derivative to be zero,

$$\frac{\partial \mathcal{L}_{pre}}{\partial \mathbf{A}_Q} := 0, \quad \frac{\partial \mathcal{L}_{pre}}{\partial \mathbf{A}_K} := 0, \quad \frac{\partial \mathcal{L}_{pre}}{\partial \mathbf{B}_Q} := 0, \quad \frac{\partial \mathcal{L}_{pre}}{\partial \mathbf{B}_K} := 0. \tag{14}$$

Solving the equations (14), we get:

$$
\begin{aligned}
\mathbf{A}_Q &= \mathbf{Q} \left( \mathbf{K}^\top \mathbf{A}_K + \lambda_{pQ}\mathbf{B}_Q^\top \right) \left( \mathbf{A}_K^\top \mathbf{A}_K + \lambda_{pQ}\mathbf{B}_Q\mathbf{B}_Q^\top \right)^{-1} \\
\mathbf{A}_K &= \mathbf{K} \left( \mathbf{Q}^\top \mathbf{A}_Q + \lambda_{pK}\mathbf{B}_K^\top \right) \left( \mathbf{A}_Q^\top \mathbf{A}_Q + \lambda_{pK}\mathbf{B}_K\mathbf{B}_K^\top \right)^{-1} \\
\mathbf{B}_Q &= (\mathbf{A}_Q^\top \mathbf{A}_Q)^{-1}\mathbf{A}_Q^\top \mathbf{Q} \\
\mathbf{B}_K &= (\mathbf{A}_K^\top \mathbf{A}_K)^{-1}\mathbf{A}_K^\top \mathbf{K}
\end{aligned}
$$

### A.2 Decode Derivatives

The Lagrangian (8) is formulated as:

$$
\begin{aligned}
\mathcal{L}_{\mathrm{dec}} = {}& \frac{1}{2} \left\| \widehat{\mathbf{q}}_t \mathbf{B}_{Q,t-1} - \mathbf{q}_t \right\|_{\mathrm{F}}^2 + \frac{1}{2} \left\| \widehat{\mathbf{k}}_t \mathbf{B}_{K,t-1} - \mathbf{k}_t \right\|_{\mathrm{F}}^2 \\
& + \frac{\lambda_{d1}}{2} \left( \widehat{\mathbf{q}}_t \widehat{\mathbf{k}}_t^\top - \mathbf{q}_t \mathbf{k}_t^\top \right)^2 + \frac{\lambda_{d2}}{2} \left\| \widehat{\mathbf{q}}_t \mathbf{A}_{K,\Omega,t-1}^\top - \mathbf{q}_t \mathbf{K}_{\Omega,t-1}^\top \right\|_{\mathrm{F}}^2.
\end{aligned}
$$

Compute partial derivative:

$$
\begin{aligned}
\frac{\partial \mathcal{L}_{\mathrm{dec}}}{\partial \widehat{\mathbf{q}}_t} &= \left( \widehat{\mathbf{q}}_t \mathbf{B}_{Q,t-1} - \mathbf{q}_t \right) \mathbf{B}_{Q,t-1}^\top \\
&\quad + \lambda_{d1} \left( \widehat{\mathbf{q}}_t \widehat{\mathbf{k}}_t^\top - \mathbf{q}_t \mathbf{k}_t^\top \right) \widehat{\mathbf{k}}_t + \lambda_{d2} \left( \widehat{\mathbf{q}}_t \mathbf{A}_{K,\Omega,t-1}^\top - \mathbf{q}_t \mathbf{K}_{\Omega,t-1}^\top \right) \mathbf{A}_{K,\Omega,t-1}. \\
\frac{\partial \mathcal{L}_{\mathrm{dec}}}{\partial \widehat{\mathbf{k}}_t} &= \left( \widehat{\mathbf{k}}_t \mathbf{B}_{K,t-1} - \mathbf{k}_t \right) \mathbf{B}_{K,t-1}^\top + \lambda_{d1} \left( \widehat{\mathbf{q}}_t \widehat{\mathbf{k}}_t^\top - \mathbf{q}_t \mathbf{k}_t^\top \right)^\top \widehat{\mathbf{q}}_t
\end{aligned}
\tag{15}
$$

By setting the partial derivative to be zero, we get:

$$\widehat{\mathbf{q}}_t \left( \mathbf{B}_{Q,t-1}\mathbf{B}_{Q,t-1}^\top + \lambda_{d1}\widehat{\mathbf{k}}_t^\top \widehat{\mathbf{k}}_t + \lambda_{d2}\mathbf{A}_{K,\Omega,t-1}^\top \mathbf{A}_{K,\Omega,t-1} \right) = \mathbf{q}_t \mathbf{B}_{Q,t-1}^\top + \lambda_{d1}\mathbf{q}_t\mathbf{k}_t^\top \widehat{\mathbf{k}}_t + \lambda_{d2}\mathbf{q}_t \mathbf{K}_{\Omega,t-1}^\top \mathbf{A}_{K,\Omega,t-1},$$

$$\widehat{\mathbf{q}}_t = \left(\mathbf{q}_t \mathbf{B}_{Q,t-1}^\top + \lambda_{d1}\mathbf{q}_t \mathbf{k}_t^\top \widehat{\mathbf{k}}_t + \lambda_{d2}\mathbf{q}_t \mathbf{K}_{\Omega,t-1}^\top \mathbf{A}_{K,\Omega,t-1}\right)\left(\mathbf{B}_{Q,t-1}\mathbf{B}_{Q,t-1}^\top + \lambda_{d1}\widehat{\mathbf{k}}_t^\top \widehat{\mathbf{k}}_t + \lambda_{d2}\mathbf{A}_{K,\Omega,t-1}^\top \mathbf{A}_{K,\Omega,t-1}\right)^{-1}.$$

Similarly, we have:

$$\widehat{\mathbf{k}}_t \mathbf{B}_{K,t-1}\mathbf{B}_{K,t-1}^\top + \lambda_{d1}\widehat{\mathbf{k}}_t\widehat{\mathbf{q}}_t^\top\widehat{\mathbf{q}}_t = \mathbf{k}_t\mathbf{B}_{K,t-1}^\top + \lambda_{d1}\mathbf{k}_t\mathbf{q}_t^\top\widehat{\mathbf{q}}_t$$

$$\widehat{\mathbf{k}}_t\left(\mathbf{B}_{K,t-1}\mathbf{B}_{K,t-1}^\top + \lambda_{d1}\widehat{\mathbf{q}}_t^\top\widehat{\mathbf{q}}_t\right) = \left(\mathbf{k}_t\mathbf{B}_{K,t-1}^\top + \lambda_{d1}\mathbf{k}_t\mathbf{q}_t^\top\widehat{\mathbf{q}}_t\right)$$

$$\widehat{\mathbf{k}}_t = \left(\mathbf{k}_t\mathbf{B}_{K,t-1}^\top + \lambda_{d1}\mathbf{k}_t\mathbf{q}_t^\top\widehat{\mathbf{q}}_t\right)\left(\mathbf{B}_{K,t-1}\mathbf{B}_{K,t-1}^\top + \lambda_{d1}\widehat{\mathbf{q}}_t^\top\widehat{\mathbf{q}}_t\right)^{-1}$$

Update $\mathbf{B}_{Q,t-1}$ and $\mathbf{B}_{K,t-1}$. First, compute the partial derivative with respect to $\mathbf{B}_{Q,t-1}$ and $\mathbf{B}_{K,t-1}$,

$$\frac{\partial \mathcal{L}_{\text{dec}}}{\partial \mathbf{B}_{Q,t-1}} = \widehat{\mathbf{q}}_t^\top \left(\widehat{\mathbf{q}}_t\mathbf{B}_{Q,t-1} - \mathbf{q}_t\right) = \widehat{\mathbf{q}}_t^\top\widehat{\mathbf{q}}_t\mathbf{B}_{Q,t-1} - \widehat{\mathbf{q}}_t^\top\mathbf{q}_t,$$

$$\frac{\partial \mathcal{L}_{\text{dec}}}{\partial \mathbf{B}_{K,t-1}} = \widehat{\mathbf{k}}_t^\top \left(\widehat{\mathbf{k}}_t\mathbf{B}_{K,t-1} - \mathbf{k}_t\right) = \widehat{\mathbf{k}}_t^\top\widehat{\mathbf{k}}_t\mathbf{B}_{K,t-1} - \widehat{\mathbf{k}}_t^\top\mathbf{k}_t.$$

Since the outer product of row vector, $\widehat{\mathbf{q}}_t^\top\widehat{\mathbf{q}}_t \in \mathbb{R}^{r\times r}$ is a rank 1 matrix, it is not invertible. Therefore, there is no closed-form solution for both $\mathbf{B}_{Q,t-1}$ and $\mathbf{B}_{K,t-1}$.

To update $\mathbf{B}_{Q,t-1}$ and $\mathbf{B}_{K,t-1}$, gradient descent is utilized. The gradient of $\mathcal{L}_{\text{dec}}$ with respect to $\mathbf{B}_{Q,t-1}$ and $\mathbf{B}_{K,t-1}$ are:

$$\begin{aligned}\nabla\mathbf{B}_{Q,t-1} &= \widehat{\mathbf{q}}_t^\top\left(\widehat{\mathbf{q}}_t\mathbf{B}_{Q,t-1} - \mathbf{q}_t\right)\\ \nabla\mathbf{B}_{K,t-1} &= \widehat{\mathbf{k}}_t^\top\left(\widehat{\mathbf{k}}_t\mathbf{B}_{K,t-1} - \mathbf{k}_t\right).\end{aligned} \tag{16}$$

The gradient descent update rules are:

$$\begin{aligned}\mathbf{B}_{Q,t} &\leftarrow \mathbf{B}_{Q,t-1} - \eta_Q\nabla\mathbf{B}_{Q,t-1},\\ \mathbf{B}_{K,t} &\leftarrow \mathbf{B}_{K,t-1} - \eta_K\nabla\mathbf{B}_{K,t-1},\end{aligned} \tag{17}$$

where $\eta_Q$ and $\eta_K$ are the learning rates for $\mathbf{B}_{Q,t-1}$ and $\mathbf{B}_{K,t-1}$, respectively. Plugging the update rule of $\mathbf{B}_{Q,t-1}$ into the Lagrangian (8), we have,

$$\arg\min_{\eta_Q} \frac{1}{2}\left\|\widehat{\mathbf{q}}_t\left(\mathbf{B}_{Q,t-1} - \eta_Q\nabla\mathbf{B}_{Q,t-1}\right) - \mathbf{q}_t\right\|_{\text{F}}^2. \tag{18}$$

This is a convex quadratic optimization problem with respect to the learning rate $\eta_Q$. Therefore, the optimal $\eta_Q$ is given by,

$$\left(\widehat{\mathbf{q}}_t\mathbf{B}_{Q,t-1} - \mathbf{q}_t - \eta_Q\widehat{\mathbf{q}}_t\nabla\mathbf{B}_{Q,t-1}\right)\left(\widehat{\mathbf{q}}_t\nabla\mathbf{B}_{Q,t-1}\right)^\top = 0. \tag{19}$$

And the optimal $\eta_Q^*$ will be,

$$\eta_Q^* = \frac{\left(\widehat{\mathbf{q}}_t\mathbf{B}_{Q,t-1} - \mathbf{q}_t\right)\left(\widehat{\mathbf{q}}_t\nabla\mathbf{B}_{Q,t-1}\right)^\top}{\left(\widehat{\mathbf{q}}_t\nabla\mathbf{B}_{Q,t-1}\right)\left(\widehat{\mathbf{q}}_t\nabla\mathbf{B}_{Q,t-1}\right)^\top}. \tag{20}$$

Similarly, for $\mathbf{B}_{K,t-1}$, the optimal $\eta_K^*$ can be computed as,

$$\eta_K^* = \frac{\left(\widehat{\mathbf{k}}_t\mathbf{B}_{K,t-1} - \mathbf{k}_t\right)\left(\widehat{\mathbf{k}}_t\nabla\mathbf{B}_{K,t-1}\right)^\top}{\left(\widehat{\mathbf{k}}_t\nabla\mathbf{B}_{K,t-1}\right)\left(\widehat{\mathbf{k}}_t\nabla\mathbf{B}_{K,t-1}\right)^\top}. \tag{21}$$

## B  Additional Results

### B.1  Results on RULER 128K And LongBench

Since the inverse computation in PyTorch operates in `float32`, the parameters are temporarily cast to `float32` before executing Algorithm 1 and Algorithm 2. The models are otherwise run with `bfloat16` for inference.

The CPU used is an AMD EPYC 7742 64-Core Processor, featuring 64 cores with 2 threads per core. For the RULER 128K experiment, the model runs on a single NVIDIA A100 GPU with 80 GB of memory. For the LongBench experiment, the model is executed on a single A100 GPU with 40 GB of memory. The batch size is set to one. Further results are provided in Table 5.

Table 5: Comparison of different models and methods on RULER 128K (left; top-2048, lite tokens=64) and LongBench (right; rank $r = 16$, lite tokens=64).

| Methods | S1 | S2 | MK1 | MK2 | MQ | MV | QA-1 | QA-2 | VT | FWE | | NQA | MQA | GRep | SAM | PRetr | LCC |
|---|---|---|---|---|---|---|---|---|---|---|---|---|---|---|---|---|---|
| Llama-3-8B-1M | 100.00 | 100.00 | 98.96 | 98.96 | 98.96 | 95.57 | 75.00 | 48.96 | 78.54 | 71.85 | | 18.98 | 41.84 | 34.18 | 35.96 | 81.50 | 56.07 |
| Loki | 18.75 | 1.04 | 2.08 | 0.00 | 1.56 | 0.78 | 4.17 | 13.54 | 26.04 | 25.35 | | 2.26 | 10.19 | 28.97 | 7.84 | 40.52 | 31.44 |
| InfiniGen | 100.00 | 98.96 | 84.38 | 53.13 | 63.28 | 54.95 | 65.63 | 48.96 | **81.67** | 50.35 | | 14.39 | 31.46 | 27.38 | 21.97 | 74.30 | 38.58 |
| Quest | 100.00 | 100.00 | **98.96** | 77.08 | 97.65 | 93.49 | 60.42 | 50.00 | 77.08 | 65.63 | | **20.13** | 36.63 | 27.11 | 35.63 | 79.00 | 53.64 |
| ShadowKV | 100.00 | 100.00 | 97.92 | **98.96** | 96.88 | 95.83 | 72.92 | 52.08 | **81.67** | **72.57** | | 17.17 | 39.73 | 31.62 | 35.87 | 80.00 | 63.93 |
| LRQK $r = 32$ | 81.00 | 100.00 | 97.00 | 42.00 | **99.25** | **98.00** | 75.00 | **53.00** | 80.20 | 69.67 | top1024 | 16.52 | **40.48** | 20.40 | 26.35 | 89.00 | 66.13 |
| LRQK $r = 16$ | 63.67 | 69.00 | 50.00 | 90.00 | 80.25 | 81.50 | 70.00 | 56.00 | 64.00 | 63.67 | top2048 | 15.86 | 40.15 | **34.15** | **36.64** | **91.50** | **66.53** |
| Llama-3.1-8B | 100.00 | 100.00 | 98.96 | 91.67 | 98.96 | 95.31 | 82.29 | 47.92 | 68.96 | 71.18 | | 31.56 | 55.10 | 34.45 | 29.84 | 100.00 | 67.31 |
| Loki | 68.75 | 32.29 | 32.29 | 20.83 | 42.71 | 28.65 | 41.67 | 33.33 | 24.79 | 29.86 | | 2.31 | 18.89 | 31.16 | 15.91 | 94.88 | 44.60 |
| InfiniGen | 100.00 | 77.08 | 78.13 | 13.54 | 58.07 | 47.40 | 65.63 | 41.67 | 60.83 | 50.35 | | 27.23 | 52.72 | 29.61 | 24.42 | 98.93 | 56.89 |
| Quest | 100.00 | 98.96 | 97.92 | 34.38 | 93.49 | 88.54 | 70.83 | 44.79 | 65.63 | **68.40** | | 29.70 | 49.04 | 30.43 | 29.85 | 98.50 | 57.35 |
| ShadowKV | 100.00 | 100.00 | 100.00 | **83.33** | 97.92 | 92.19 | **81.25** | **48.96** | 67.08 | 64.93 | | 30.93 | 55.20 | **32.79** | **30.40** | **99.50** | **66.03** |
| LRQK $r = 32$ | 87.00 | 100.00 | 98.00 | 80.00 | **98.25** | **96.25** | 81.00 | 41.00 | **70.00** | 44.67 | top1024 | 29.01 | **55.85** | 19.50 | 12.02 | **99.50** | 24.60 |

## B.2 Results on Llama And Qwen

More results of the proposed methods on two different models, Llama-3-8B-1M and Qwen2.5-7B, are shown in Table 6. The model is running on a single A100 GPU with 40G memory. The configuration for LRQK are: rank=16, top-256 active tokens, and 16 lite tokens.

Table 6: More results on subset of RULER-4K/8K/16K of two models.

| RULER 4K | QA-1 | QA-2 | VT | RULER 4K | QA-1 | QA-2 | VT |
|---|---|---|---|---|---|---|---|
| LLaMA-3-8B-1M | 82.00 | 58.00 | 99.00 | Qwen2.5-7B | 90.00 | 64.00 | 99.20 |
| + LRQK | 84.00 | 57.00 | 98.80 | + LRQK | 91.00 | 65.00 | 96.60 |
| RULER 8K | QA-1 | QA-2 | VT | RULER 8K | QA-1 | QA-2 | VT |
| LLaMA-3-8B-1M | 82.00 | 57.00 | 99.60 | Qwen2.5-7B | 82.00 | 57.00 | 97.40 |
| + LRQK | 79.00 | 47.00 | 94.00 | + LRQK | 78.00 | 56.00 | 70.00 |
| RULER 16K | QA-1 | QA-2 | VT | RULER 16K | QA-1 | QA-2 | VT |
| LLaMA-3-8B-1M | 80.00 | 57.00 | 99.80 | Qwen2.5-7B | 76.00 | 63.00 | 98.80 |
| + LRQK | 77.00 | 52.00 | 57.20 | + LRQK | 71.00 | 62.00 | 64.20 |

## B.3 Results on Mistral and Phi-3

Two additional models, 'mistralai/Mistral-7B-Instruct-v0.3' (Mistral) and 'microsoft/Phi-3-mini-128k-instruct' (Phi-3), are evaluated under the default LRQK hyperparameters (rank=32, top-2048, and 64 lite tokens). Due to the 48 GB memory constraints of the NVIDIA A6000 GPU, Mistral is evaluated on RULER 32K and Phi-3-mini is evaluated on RULER 16K. Table 7 summarizes the results, demonstrating that LRQK generalizes effectively to architecturally diverse models.

## B.4 Results on Larger Models

To evaluate the scalability of LRQK to larger models, additional experiments are conducted on 'Qwen/Qwen2.5-14B-Instruct'(Qwen 14B) with 64K context length and 'Qwen/Qwen2.5-32B-Instruct' (Qwen 32B) with 16K context length under A100 80G GPU. Following the official guidance from Qwen2.5, YaRN [29] is applied for positional encoding extension in the 64K context setting. All experiments use the default LRQK hyperparameters (rank=32, top-2048, and 64 lite tokens).

Table 8 presents the results on the RULER benchmark. For Qwen 14B at 64K context, LRQK yields substantial improvements on retrieval tasks, demonstrating its effectiveness at extended context lengths. For Qwen 32B at 16K context, the baseline model already achieves near-perfect performance on most tasks, with LRQK maintaining comparable results, indicating that the method does not degrade performance on models with strong native long-context capabilities.

Table 7: Evaluation results of Mistral and Phi-3 on RULER benchmarks with varying context lengths.

| RULER 32K | Mistral | +LRQK | RULER 16K | Phi-3 | +LRQK |
|---|---|---|---|---|---|
| FWE | 67.00 | 93.00 | FWE | 92.00 | 91.00 |
| S1 | 98.00 | 97.00 | S1 | 100.00 | 100.00 |
| S2 | 91.00 | 100.00 | S2 | 100.00 | 100.00 |
| MK1 | 83.00 | 97.00 | MK1 | 96.00 | 96.00 |
| MK2 | 63.00 | 51.00 | MK2 | 100.00 | 100.00 |
| MV | 85.75 | 95.00 | MV | 90.00 | 90.25 |
| MQ | 83.75 | 93.00 | MQ | 90.50 | 87.50 |
| QA-1 | 59.00 | 63.00 | QA-1 | 80.00 | 81.00 |
| QA-2 | 45.00 | 47.00 | QA-2 | 51.00 | 51.00 |
| VT | 98.40 | 96.40 | VT | 99.60 | 99.60 |

Table 8: Performance comparison on Qwen2.5-14B-Instruct (64K context) and Qwen2.5-32B-Instruct (16K context) with and without LRQK on the RULER benchmark. Columns '+LRQK' indicate the performance of the model with LRQK.

| RULER 64K | Qwen 14B | +LRQK | RULER 16K | Qwen 32B | +LRQK |
|---|---|---|---|---|---|
| FWE | 90.33 | 78.33 | FWE | 94.67 | 94.00 |
| S1 | 57.00 | 99.00 | S1 | 100.00 | 100.00 |
| S2 | 55.00 | 96.00 | S2 | 100.00 | 100.00 |
| MK1 | 46.00 | 80.00 | MK1 | 100.00 | 100.00 |
| MK2 | 24.00 | 22.00 | MK2 | 99.00 | 99.00 |
| MV | 54.00 | 82.25 | MV | 100.00 | 100.00 |
| MQ | 49.75 | 88.50 | MQ | 100.00 | 100.00 |
| QA-1 | 64.00 | 50.00 | QA-1 | 86.00 | 86.00 |
| QA-2 | 41.00 | 36.00 | QA-2 | 66.00 | 66.00 |
| VT | 57.80 | 97.20 | VT | 86.40 | 82.60 |

## B.5 Results With KVQuant

To investigate whether LRQK is compatible with quantization based compression methods, the combination of LRQK with KVQuant [6], a representative KV cache quantization approach, is evaluated. Experiments are conducted on 'meta-llama/Llama-3.1-8B-Instruct' (Llama-3.1-8B) and 'Qwen/Qwen2.5-7B-Instruct' (Qwen 2.5-7B-Instruct) using the RULER 32K benchmark. The LRQK hyperparameters are set to $r = 32$, top-$k = 2048$, and 64 lite tokens.

Table 9 presents the results comparing models with KVQuant alone versus the combination of KVQuant and LRQK. For Llama-3.1-8B, the combined approach maintains competitive performance across most tasks. For Qwen2.5-7B, combining both methods yields mixed results: while some tasks show improvements (e.g., MK1, MV, MQ), others exhibit slight degradation (e.g., MK2). Overall, the results demonstrate that LRQK can be effectively combined with quantization methods without catastrophic performance loss, suggesting that the proposed method and quantization operate on complementary aspects of KV cache compression.

## B.6 Impact of Initialization Strategies

Three initialization strategies are investigated for the low rank factors $\mathbf{A}_Q \in \mathbb{R}^{l \times r}$ and $\mathbf{A}_K \in \mathbb{R}^{l \times r}$, where $l$ is the sequence length, and $r$ is the chosen rank, and the batch size or the number of attention heads are ignored as discussed in Section 4. The design rationale for these strategies stems from the goal of finding effective initialization that either minimize reconstruction error or provide sufficient representational capacity for the low rank approximation.

**Random Gaussian Initialization (randn).** The first strategy initializes both factors with random Gaussian noise:

$$\mathbf{A}_Q, \mathbf{A}_K \sim \mathcal{N}(0, 1). \tag{22}$$

Table 9: Performance comparison on RULER 32K with KVQuant applied alone and in combination with LRQK. Columns '+KVQuant' indicate KVQuant only, and columns '++LRQK' indicate both KVQuant and LRQK applied.

| RULER 32K | Llama-3.1-8B | | Qwen2.5-7B-Instruct | |
|---|---|---|---|---|
| | +KVQuant | ++LRQK | +KVQuant | ++LRQK |
| FWE | 89.58 | 85.00 | 94.00 | 90.33 |
| S1 | 100.00 | 100.00 | 100.00 | 100.00 |
| S2 | 100.00 | 100.00 | 100.00 | 99.00 |
| MK1 | 100.00 | 100.00 | 96.00 | 100.00 |
| MK2 | 95.00 | 96.00 | 76.00 | 56.00 |
| MV | 95.31 | 96.75 | 70.50 | 93.50 |
| MQ | 89.00 | 89.00 | 97.00 | 99.25 |
| QA-1 | 83.00 | 82.00 | 73.00 | 74.00 |
| QA-2 | 56.25 | 51.00 | 52.00 | 53.00 |
| VT | 87.50 | 95.40 | 95.40 | 90.60 |

This approach provides a simple, parameter-free initialization that avoids any dependence on the input matrices. While it does not leverage information from $\mathbf{Q}$ and $\mathbf{K}$, it offers computational efficiency and serves as a neutral baseline.

**Independent Top-$r$ Selection (top).** The second strategy aims to minimize reconstruction error by selecting the most salient dimensions from $\mathbf{Q}$ and $\mathbf{K}$ independently. The $L_1$ norm is computed of each dimension across the sequence:

$$\mathbf{s}_Q = \sum_{i=1}^{l} |\mathbf{Q}[i,:]| \in \mathbb{R}^{1 \times d}, \qquad \mathbf{s}_K = \sum_{i=1}^{l} |\mathbf{K}[i,:]| \in \mathbb{R}^{1 \times d}, \tag{23}$$

where $d$ is the head dimension. With these importance scores, the initialized low rank factors are constructed as:

$$\mathbf{A}_Q \leftarrow \mathbf{Q}_{\text{top-}r(\mathbf{s}_Q)}, \qquad \mathbf{A}_K \leftarrow \mathbf{K}_{\text{top-}r(\mathbf{s}_K)}. \tag{24}$$

This strategy independently optimizes each factor to capture the most significant features of its respective matrix.

**Joint Top-$r$ Selection (topcol).** The third strategy selects dimensions based on the combined importance in both $\mathbf{Q}$ and $\mathbf{K}$, ensuring that $\mathbf{A}_Q$ and $\mathbf{A}_K$ share the same column indices:

$$\mathbf{s}_{QK} = \mathbf{s}_Q + \mathbf{s}_K, \qquad \Omega = \text{top-}r(\mathbf{s}_{QK}), \tag{25}$$

$$\mathbf{A}_Q \leftarrow \mathbf{Q}_\Omega, \qquad \mathbf{A}_K \leftarrow \mathbf{K}_\Omega. \tag{26}$$

This joint selection promotes alignment between the query and key subspaces, potentially facilitating more coherent attention patterns in the low rank approximation.

**Experimental Setup and Results.** These three initialization strategies are evaluated on 'meta-llama/Llama-3.1-8B-Instruct-1M' and 'Qwen/Qwen2.5-7B-Instruct' using the RULER 16K benchmark with default LRQK hyperparameters ($r = 32$, top-$k = 2048$, 64 lite tokens).

Table 10 presents the results. Remarkably, all three initialization strategies achieve nearly identical performance across most tasks, with differences typically within 1-2 percentage points. This robustness suggests that the specific choice of initialization has minimal impact on the final performance, indicating that the Algorithm 1 is resilient to the initialization of the low rank factors. The consistency across strategies, from uninformed random initialization to carefully selected subspaces, demonstrates the method's inherent stability.

Given this empirical equivalence, the **randn** initialization is recommended as a default due to its computational efficiency: it avoids the overhead of computing importance scores and performing top-$k$ selection, making it particularly suitable for large-scale deployments.

Table 10: Performance comparison of initialization strategies for $\mathbf{A}_Q$ and $\mathbf{A}_K$ on RULER 16K. Despite their different design rationales, all three methods achieve comparable results.

| RULER 16K | Llama-3.1-8B-1M | | | Qwen2.5-7B-Instruct | | |
|---|---|---|---|---|---|---|
| | randn | top | topcol | randn | top | topcol |
| FWE | 86.67 | 86.67 | 87.00 | 85.67 | 86.33 | 86.33 |
| S1 | 100.00 | 100.00 | 100.00 | 100.00 | 100.00 | 100.00 |
| S2 | 100.00 | 100.00 | 100.00 | 100.00 | 100.00 | 100.00 |
| MK1 | 100.00 | 100.00 | 100.00 | 99.00 | 99.00 | 99.00 |
| MK2 | 99.00 | 99.00 | 99.00 | 91.00 | 91.00 | 91.00 |
| MV | 98.75 | 98.50 | 98.50 | 97.75 | 98.00 | 97.75 |
| MQ | 94.75 | 94.50 | 94.75 | 99.75 | 99.75 | 99.75 |
| QA-1 | 89.00 | 89.00 | 89.00 | 76.00 | 76.00 | 76.00 |
| QA-2 | 61.00 | 61.00 | 61.00 | 61.00 | 62.00 | 62.00 |
| VT | 83.80 | 83.00 | 85.40 | 98.80 | 98.80 | 98.40 |

## C  Runtime Performance Analysis

### C.1  Throughput Results

Runtime performances of LRQK are evaluated on 'meta-llama/Llama-3.1-8B-Instruct-1M', comparing it against standard GPU-only and CPU offloading approaches. Experiments are conducted using the Hugging Face `transformers` library[2] on a single NVIDIA A100 GPU (40 GB memory) with batch size 1. We utilize the RULER QA-2 (HotpotQA) dataset with context lengths ranging from 4K to 64K tokens. The parameters of LRQK are set as $r = 16$, top-$k = 1024$ active tokens, and 16 lite tokens.

Experimental Configurations:

- **GPU only**: All KV caches stored in GPU memory (baseline).
- **CPU offload**: KV caches offloaded to CPU memory and transferred to GPU during decoding.
- **LRQK default**: Full LRQK implementation with hit/miss buffer enabled for optimized cache management.
- **LRQK no hit/miss**: LRQK without hit/miss buffer to isolate the impact of buffer optimization.

Table 11 presents the tokens processed per second during prefill (P) and decode (D) stages. The GPU-only method achieves the highest throughput for shorter contexts (4K to 32K) but encounters out-of-memory (OOM) errors at 64K tokens due to the growth of KV cache size.

CPU offload enables processing of longer contexts but suffers severe throughput degradation, particularly during decoding at 32K to 64K tokens. This performance drop stems from the need to transfer the entire KV cache between CPU and GPU for each decoding step, creating a bottleneck that scales linearly with context length.

In contrast, LRQK default maintains relatively stable decoding throughput across all context lengths, demonstrating consistent performance even at 64K tokens. This stability arises from LRQK's selective transfer mechanism: instead of moving the entire KV cache, only the top-$k$ active tokens and lite tokens are transferred from CPU to GPU, significantly reducing data movement overhead. At 64K context, LRQK achieves faster decoding than CPU offload while avoiding the OOM issues of GPU-only execution.

The LRQK no hit/miss configuration shows reduced decoding performance compared to LRQK default, highlighting the importance of the hit/miss buffer optimization. Without this buffer, the method incurs additional overhead from repeated top-$k$ selection operations, which are not well-optimized for CPU cache locality. Notably, prefill performance remains comparable across LRQK variants, as this stage is less sensitive to cache management strategies.

---

[2]https://huggingface.co/docs/transformers/v4.52.2/en/main_classes/text_generation

Table 11: Throughput comparison (tokens/s) for 'meta-llama/Llama-3.1-8B-Instruct-1M' on RULER QA-2 across context lengths. P: prefill stage; D: decode stage. LRQK default maintains stable decode performance while avoiding OOM at 64K tokens.

|  | 4K | | 8K | | 16K | | 32K | | 64K | |
|---|---|---|---|---|---|---|---|---|---|---|
|  | P | D | P | D | P | D | P | D | P | D |
| GPU only | 37500.29 | 35.40 | 37734.19 | 35.64 | 33649.79 | 35.36 | 32111.45 | 35.77 | OOM | OOM |
| CPU offload | 6945.10 | 4.31 | 6984.35 | 4.32 | 7073.53 | 2.40 | 4849.19 | 0.98 | 4131.00 | 0.50 |
| LRQK default | 7120.49 | 5.68 | 7280.39 | 5.72 | 6379.91 | 5.49 | 5436.66 | 5.36 | 4103.05 | 6.80 |
| LRQK no hit/miss | 7180.64 | 2.50 | 6747.19 | 2.53 | 5317.06 | 2.31 | 4920.02 | 2.48 | 4163.86 | 1.95 |

## C.2 Comparison with Baseline Methods

To provide a more comprehensive performance analysis, LRQK is compared against vanilla attention and ShadowKV[16] on a text summarization task using 20 samples from LongBench with 32K context and up to 128 output tokens. Method ShadowKV is configured with its default parameters (sparse budget = 2048, rank=160, chunk size=8), and the definition of 'rank' is not the same as this paper, please refer to the original paper for more details. Experiments are conducted on NVIDIA GeForce RTX 3090 (24 GB) GPUs. Due to memory constraints, vanilla attention requires 2 GPUs, while ShadowKV and LRQK operate on a single GPU.

Table 12 presents detailed performance metrics. Vanilla attention achieves the highest throughput but requires dual-GPU deployment with large memory usage across both two devices. ShadowKV reduces throughput compared to vanilla. LRQK variants achieve 489 to 646 tokens/s depending on hyperparameters, representing a trade-off between memory efficiency and speed. Basically, LRQK with less rank and smaller $k$ can achieve larger tokens/s. The GPU power of LRQK is smaller than Vanilla and ShadowKV, which means the proposed method is not fully utilizing the GPU resources. This is one limitation of the proposed method. It requires more time to wait data indexing in CPU.

Table 12: Performance comparison on NVIDIA GeForce RTX 3090 (24 GB, 250W) for text summarization (LongBench, 32K context). Vanilla attention uses 2 GPUs; other methods use 1 GPU. Tokens/s is computed via (number of all tokens / total time).

|  | Vanilla | | ShadowKV | LRQK | | | |
|---|---|---|---|---|---|---|---|
|  | | | | $k = 2048$ | | $k = 1024$ | |
|  | GPU0 | GPU1 | | $r = 32$ | $r = 16$ | $r = 32$ | $r = 16$ |
| Average GPU Util (%) | 38.61 | 48.38 | 58.00 | 63.64 | 65.29 | 56.86 | 51.15 |
| Max GPU Util (%) | 99.95 | 100.00 | 100.00 | 100.00 | 100.00 | 100.00 | 100.00 |
| GPU Memory (GB) | 16.94 | 17.56 | 17.93 | 19.82 | 18.85 | 19.51 | 19.33 |
| GPU Power (W) | 187.24 | 233.21 | 243.85 | 203.98 | 211.96 | 199.59 | 199.85 |
| Tokens/s | 1775.70 | | 1083.55 | 489.39 | 556.22 | 603.05 | 646.33 |
| Time (s) | 17.87 | | 29.29 | 64.84 | 57.04 | 52.62 | 57.04 |

# D Hyperparameter Selection Guidelines

LRQK introduces several hyperparameters that require configuration: rank $r$ for low rank approximation, top-$k$ for active token selection, number of lite tokens, and convergence parameters (iterations and tolerance). While optimal values vary across model architectures and tasks, practical guidelines are provided based on our extensive experiments to minimize tuning effort.

**Default Configuration.** It is recommended starting with the following default configuration, which achieves strong performance across diverse settings: rank $r = 32$, active tokens top-$k = 2048$, lite tokens 64, iterations 2, and tolerance $10^{-2}$. This configuration provides a balanced trade-off between approximation quality, memory efficiency, and computational overhead, and serves as an effective starting point for task-specific tuning.

**Rank $r$.** The rank $r$ controls the dimensionality of the low rank factors $\mathbf{A}_Q, \mathbf{A}_K \in \mathbb{R}^{\cdots l \times r}$ for each attention head independently. Since typical attention heads have dimension $d_{\text{head}} = 128$, the theoretical range for $r$ is $[1, 128]$. In practice, it is found that $r \in \{8, 16, 32, 48\}$ can provide a favorable balance between approximation quality and computational cost. Lower ranks (e.g., $r = 8$) reduce memory usage but may sacrifice accuracy, while higher ranks (e.g., $r = 48$) improve approximation at the cost of increased computation. For most applications, $r = 32$ offers an effective compromise.

**Top-$k$ Active Tokens.** The top-$k$ parameter determines how many high-attention tokens are retained in GPU memory and directly influences the memory-context length trade-off. The optimal $k$ scales with context length: short contexts ($\leq$ 4K) $k = 256$ suffices for most tasks, medium contexts (8K to 16K): $k = 512$ to $1024$ may provide good coverage, long contexts ($\geq$ 32K) $k = 2048$ is recommended. Smaller $k$ values reduce memory footprint but may miss important context, while larger $k$ improves recall at the cost of increased GPU memory usage and data transfer overhead. The relationship between $k$ and context length should be considered when deploying LRQK for production use cases.

**Lite Tokens.** The number of lite tokens specifies how many recent tokens are always retained in GPU memory to capture local context. Recommend choices are $\{16, 32, 64\}$, with minimal impact on overall computational cost. This parameter is relatively insensitive; it is recommended $64$ lite tokens as a conservative default that ensures sufficient local context coverage.

**Convergence Parameters.** The iterative updates in Algorithms 1 and 2 require two convergence parameters:

- **Iterations**: Number of update cycles for low rank factors. Typically, 2 or 3 iterations may suffice, as the approximation would converge.

- **Tolerance**: Convergence threshold defined as the mean squared error between successive low rank matrices. Values of $10^{-2}$ or $10^{-3}$ work well in practice. Tighter tolerance (e.g., $10^{-3}$) improves accuracy slightly but increases computation.

**Tuning Strategy.** The search space for LRQK hyperparameters is relatively compact compared to other optimization based methods. It is recommended the following tuning protocol:

1. Start with the default configuration ($r = 32, k = 2048$, 64 lite tokens, 2 iterations, tolerance $10^{-2}$).

2. If memory is limited, reduce $k$ proportionally to the context length or decrease $r$.

3. If accuracy is insufficient, increase $k$ (up to hardware limits) or $r$.

In our experiments, this strategy typically requires evaluating fewer configurations to identify near-optimal settings for a given task and model architecture.

**Implementation and Integration Overhead.** LRQK introduces modest computational overhead relative to standard attention:

- **Prefill stage**: Additional cost for computing low rank projections, partially offset by reduced attention computation over selected tokens.

- **Decode stage**: Incremental updates to low rank matrices add computation, but attention over fewer tokens ($k$ instead of full sequence) reduces overall cost.

- **Cache management**: CPU-GPU transfers for selected tokens and cache lookup operations add latency, though this is mitigated by transferring only $k$ tokens rather than the entire KV cache.

