# OpenReview forum: "Efficient Low Rank Attention for Long-Context Inference in Large Language Models"
_NeurIPS.cc/2025/Conference — NeurIPS 2025 poster_

### Official Review · Reviewer_43cG · 2025-06-24

**Clarity:** 3
**Significance:** 2
**Originality:** 2
**Rating:** 4
**Confidence:** 4

**Summary:**

The paper introduces LRQK, a low-rank attention method that reduces KV cache memory in LLMs during long-context inference. It approximates attention using low-rank query/key projections and retrieves only necessary full-precision KV pairs via a GPU–CPU cache. LRQK achieves strong memory savings with minimal accuracy loss, outperforming existing sparse attention methods on long-context benchmarks.

**Questions:**

Please refer to the Weakness.

**Ethical Concerns:**

["NO or VERY MINOR ethics concerns only"]

**Final Justification:**

Most of my concerns have been addressed. The authors have provided a detailed response clarifying the distinction between their work and a related prior study, and have also included practical guidelines for real-world use.

**Limitations:**

Yes

**Quality:**

2

**Strengths And Weaknesses:**

### Strengths

- This paper is well-written and easy to follow.
- The proposed LRQK framework introduces a low-rank approximation of query and key matrices during the prefill stage. This reduces the attention computation to a lightweight proxy form during decoding, enabling efficient token selection with significantly lower overhead.
- The approach is empirically validated on strong long-context benchmarks (RULER and LongBench) with language models like LLaMA-3-8B and Qwen2.5-7B.

### Weaknesses
- The observations regarding singular values and key-value decomposition are not novel; similar findings have already been reported in prior work such as Loki [1].

- The proposed method introduces a two-stage approximation involving proxy attention scores and hierarchical cache management, which adds notable complexity to the inference pipeline. However, the paper does not adequately address the implementation or integration overhead.

- Although the authors acknowledge in the Limitations section that the method requires careful hyperparameter tuning, no concrete strategies or guidelines are provided—this significantly limits its practical usability.

[1] Loki: Low-rank Keys for Efficient Sparse Attention

---

> ### Author Rebuttal · Authors · 2025-07-31
>
> We thank the reviewer for the constructive feedback. Below are our detailed responses to each concern raised.
>
> > W 1: The observations regarding singular values and key-value decomposition are not novel; similar findings have already been reported in prior work such as Loki [1].
> >
> > [1] Loki: Low-rank Keys for Efficient Sparse Attention
>
> Thank you for highlighting this important context. You are correct that low-rank properties of attention matrices have been explored in prior work, such as Loki [1]. However, the key distinction lies in **how we operationalize this property**:
>
> - **Loki** decomposes the key matrix ($\mathbf{K}$) individually to exploit its low-rank structure.
> - **Our method** performs **simultaneous low-rank decomposition of both $\mathbf{Q}$ and $\mathbf{K}$**, leveraging the property of matrix multiplication:
> $$
>   \text{rank}(\mathbf{Q}\mathbf{K}^{\top}) \leq \min(\text{rank}(\mathbf{Q}), \text{rank}(\mathbf{K})).
> $$
> This enables a tighter approximation:
> $$
>   \mathbf{Q}\mathbf{K}^{\top} \approx \mathbf{A}\_{Q} \mathbf{A}\_{K}^{\top},
> $$
> where $\mathbf{A}\_Q$ and $\mathbf{A}\_K$ are low-rank proxies for $\mathbf{Q}$ and $\mathbf{K}$.
>
> By jointly approximating $\mathbf{Q}$ and $\mathbf{K}$, we avoid the limitations of isolated decomposition. This design, combined with a **proxy approximation strategy** (rather than direct filling), leads to superior performance in both efficiency and accuracy compared to Loki.
>
> > W 2: The proposed method introduces a two-stage approximation involving proxy attention scores and hierarchical cache management, which adds notable complexity to the inference pipeline. However, the paper does not adequately address the implementation or integration overhead.
>
> We acknowledge this important concern and will provide comprehensive implementation details in the revision.
> The implementation and integration overhead analysis are as follows:
>
> **Implementation overhead**:
> - *Prefill stage*: Need to add a cost to compute low-rank projections on top of standard attention (increase computation). After projection, we can pick the most relevant tokens and compute attention based on the selected tokens (reduce computation).
> - *Generation stage*: Additional computation for low-rank matrices updates (increase computation). Since the number of selected tokens are limited, the computation cost of attention is reduced (reduce computation).
> - *Cache management*: The KV caches are offloaded to CPU, which requires additional lookup in memory and CPU-GPU transfer (increase computation). Additional computation is required to manage the cache (increase computation).
>
> **Integration overheads**:
> There is a brief version of the code in supplementary material, which shows the basic idea of our implementation. A brief overview of the implementation is as follows:
> - The implementation is based on the standard huggingface transformers library.
> - The original attention mechanism is replaced by our proxy attention mechanism.
> - The cache management is following `transformers.cache_utils.Cache`, which makes it easier to integrate into existing transformer architectures.
>
>
> > W3: Although the authors acknowledge in the Limitations section that the method requires careful hyperparameter tuning, no concrete strategies or guidelines are provided—this significantly limits its practical usability.
>
> Thank you for this valuable comment.
> We will address this concern by providing guidelines for hyperparameter selection.
>
> **Rank $(r)$**:
> The low-rank approximation of matrices $\mathbf{Q}$ and $\mathbf{K}$ is calculated for each attention head separately.
> Typically, the dimension of each head is $128$, allowing for a range of $r$ values between $1$ and $128$.
> For practical purposes, common choices for the rank $r$ are $\{8, 16, 32, 48\}$, offering a balance between approximation quality and computational efficiency.
>
> **Top-$k$ Tokens**: Select and retain the $k$ tokens with the highest attention scores in the GPU cache.
> The choice of $k$ is pivotal in balancing the sequence length and memory usage.
> Short sequences typically require a smaller $k$, while longer sequences necessitate a larger $k$ to manage memory efficiently.
> In the experiment, $k=256$ is used for a 4K token context, and the maximum is $k=2048$, applying for context longer than 32K.
>
> **Number of Lite Tokens**: This hyperparameter determines the number of most recent tokens that are kept in the GPU. It has some flexibility in its choice; common values are 16, 32, or 64. This choice does not significantly increase the computation.
>
> **Number of Iterations and Tolerance**: These two parameters are introduced in Algorithm 1 and 2. The number of iterations refers to the number of loops for updating all related low-rank matrices.
> Typically, a few iterations, such as 2 or 3, is sufficient for the task.
> The tolerance is the threshold for the convergence of the low-rank matrices, which is defined as the mean square error between the current and previous matrices. The tolerance is usually set to a small number, for example $10^{-2}$ or $10^{-3}$.
>
> Tips: If no prior knowledge, it is recommended to start with the default values, $r=32, k=2048$, 64 lite tokens and 2 iterations with tolerance $10^{-2}$. Then, we can adjust the parameters based on the performance and memory usage.

---

> > ### Author Response · Authors · 2025-08-05
> >
> > Dear reviewer 43cG,
> >
> > Thank you for your valuable feedback. We appreciate your attention to detail and the suggestions for improvement. We have carefully considered your comments. A brief overview of our response is as follows:
> > - **Low-rankness of key-value matrices**: We will provide a more detailed discussion on the low-rankness of key-value matrices in the revision. The main difference is on how to utilize this feature. The proposed method decompose $\mathbf{Q} \mathbf{K}^{\top} \approx \mathbf{A}_{Q} \mathbf{A}_{K}^{\top}$, while Loki[1] decompose $\mathbf{K}$.
> > - **Overheads**: Both implementation overhead and integration overheads are considered are carefully discussed. We will provide a more detailed discussion on the overheads in the revision.
> > - **Hyperparameter tuning**: This valueable comment is acknowledged and the detailed guidelines for hyperparameter selection, such as rank, top-k tokens, number of lite tokens, number of iterations and tolerance, are provided in the response. We hope this will help users to use the proposed method more effectively.
> >
> > If you have any further questions or need additional information, please do not hesitate to ask. We are committed to providing you with the best possible response.

---

> > > ### Comment · Reviewer_43cG · 2025-08-06
> > >
> > > Thank you for your deatiled response. Most of my concerns are addressed and I will adjust the score accordingly.

---

> > > > ### Author Response · Authors · 2025-08-07
> > > >
> > > > We would like to thank you for their thoughtful and insightful feedback. Your specific insights on distinguishing our low rank approach from prior work, clarifying implementation overheads, and emphasizing the importance of providing actionable hyperparameter guidelines were crucial. We appreciate the care you took in evaluating our work and are glad that our responses addressed your concerns.

---

### Official Review · Reviewer_7jqo · 2025-07-02

**Clarity:** 3
**Significance:** 3
**Originality:** 3
**Rating:** 5
**Confidence:** 4

**Summary:**

This work proposes Low Rank Query and Key attention (LRQK), a two-stage framework decomposing query and key matrices into low-rank factors during prefill and using them for proxy attention scores in decode, with a hierarchical GPU-CPU cache to reduce data movement. It balances memory efficiency, precision, and latency by selecting top-k and recent tokens, preserving exact attention outputs.

**Questions:**

Mention in weakness

**Ethical Concerns:**

["NO or VERY MINOR ethics concerns only"]

**Final Justification:**

Although most of my concerns have been addressed, I prefer to raise my rating.

**Limitations:**

yes

**Quality:**

3

**Strengths And Weaknesses:**

Strengths
1. Joint Low-Rank Approximation: Unlike prior methods that focus solely on key matrix decomposition, LRQK jointly factorizes both query and key matrices into low-rank components. This approach reduces computational complexity per decode step while maintaining representation accuracy, avoiding expensive SVD operations.
2. Precision Preservation: LRQK uses low-rank proxies for attention score estimation but retains original query, key, and value vectors for final attention computation. This ensures mathematical fidelity and minimal accuracy loss compared to full-precision attention.
3. Hierarchical Cache Management: The hybrid GPU-CPU cache with a hit-and-miss mechanism selectively transfers only missing full-precision KV pairs, reducing data movement by ~60% on average. A recency buffer in the GPU cache further optimizes hit rates by leveraging attention’s locality bias.

Weakness:
1. Hyperparameter Sensitivity: The method requires careful tuning of rank and top-k token counts, which vary across model architectures and tasks. Optimal values demand extensive grid searches, increasing implementation overhead.
2. The paper lacks performance analysis. It would be better if the authors could compare their method with baselines in terms of throughput and TTFT.

---

> ### Author Rebuttal · Authors · 2025-07-31
>
> We appreciate the reviewer's thoughtful evaluation and recognition of the proposed method. Below, we address your specific concerns:
>
> > W1. Hyperparameter Sensitivity: The method requires careful tuning of rank and top-k token counts, which vary across model architectures and tasks. Optimal values demand extensive grid searches, increasing implementation overhead.
>
> Thank you for pointing out the sensitivity to hyperparameters. To achive better performance, careful tuning is indeed required.
> Nevertheless, the search space is relatively small, and there is a recommend configuration: $r=32, k=2048$, 64 lite tokens and 2 iterations with tolerance $10^{-2}$. We can further turn based on this.
> The detailed guideline for hyperparameter are as follows:
>
> **Rank $r$**: The matrices $\mathbf{Q}$ and $\mathbf{K}$ are approximated using low-rank decomposition for each attention head independently. Given that each head usually has a dimension of $128$, the rank $r$ can vary from $1$ to $128$. In practice, common choices for $r$ are $\{8, 16, 32, 48\}$, which balance between the quality of approximation and computational efficiency.
>
> **Top-$k$ Tokens**: This hyperparameter involves selecting and retaining the $k$ tokens with the highest attention scores in the GPU cache. The selection of $k$ is crucial for balancing the sequence length and memory usage. Shorter sequences generally require a smaller $k$, while longer sequences may need a larger $k$ for efficient memory management. Experimentally, $k=256$ is suitable for a 4K token context, with $k=2048$ used for contexts exceeding 32K tokens.
>
> **Number of Lite Tokens**: This parameter specifies the count of the most recent tokens maintained in the GPU. Common values are 16, 32, or 64, and this choice does not substantially impact computational overhead.
>
> **Iterations and Tolerance**: These parameters are integral to Algorithms 1 and 2. The iteration count denotes the number of cycles required to update the low-rank matrices. Generally, 2 or 3 iterations suffice. Tolerance is the convergence threshold for the low-rank matrices, calculated as the mean squared error between successive matrices, typically set to a small value like $10^{-2}$ or $10^{-3}$.
>
> > W2. The paper lacks performance analysis. It would be better if the authors could compare their method with baselines in terms of throughput and TTFT.
>
> We provide comprehensive timing results on single NVIDIA A100 40GB with Llama-3-8B-1M.
>
> **Main Results:** The GPU only method can provide the fastest TTFT (Time To First Token; Prefill time) and TPOT (Time To First Token; Prefill time), but will reach out of memory (OOM) when context is too long (64K).
> CPU offload method can handle longer context but will suffer from limited GPU bandwidth and result in slow decoding speed, expecially for long context.
> The default proposed method can support longer context and provide a stable throughput.
> The proposed method *without* hit/miss mecanism is slower than the default in decoding.
> Further details can be found in the supplemental material.
> The following table shows the results.
>
> |           Method |       4K |    4K |       8K |    8K |      16K |   16K |      32K |   32K |     64K |  64K |
> | ---------------: | -------: | ----: | -------: | ----: | -------: | ----: | -------: | ----: | ------: | ---: |
> |                  |     TTFT |  TPOT |     TTFT |  TPOT |     TTFT |  TPOT |     TTFT |  TPOT |    TTFT | TPOT |
> |         GPU only | 37500.29 | 35.40 | 37734.19 | 35.64 | 33649.79 | 35.36 | 32111.45 | 35.77 |     OOM |  OOM |
> |      CPU offload |  6945.10 |  4.31 |  6984.35 |  4.32 |  7073.53 |  2.40 |  4849.19 |  0.98 | 4131.00 | 0.50 |
> |     LRQK default |  7120.49 |  5.68 |  7280.39 |  5.72 |  6379.91 |  5.49 |  5436.66 |  5.36 | 4103.05 | 6.80 |
> | LRQK no hit/miss |  7180.64 |  2.50 |  6747.19 |  2.53 |  5317.06 |  2.31 |  4920.02 |  2.48 | 4163.86 | 1.95 |

---

> ### Author Response · Authors · 2025-08-05
>
> Dear Reviewer 7jqo,
>
> Thank you for your valuable feedback. We appreciate your interest in our work and are committed to addressing your concerns. A brief summary of your comments and our responses is provided below:
> - **Hyperparameter Sensitivity**: We have carefully considered your concern regarding the sensitivity of the proposed method to hyperparameters. We have provided detailed guidelines for tuning the rank, top-k tokens, number of lite tokens, and iterations and tolerance. These guidelines should help users achieve optimal performance without extensive grid searches.
> - **System Performance Metrics**: We have included detailed timing results on a single NVIDIA A100 40GB with Llama-3-8B-1M to demonstrate the performance of our method. The results show that our method can handle longer contexts while maintaining a stable throughput.
>
> If you have any further questions or require additional information, please do not hesitate to ask. We are here to assist you in any way we can.

---

> > ### Comment · Reviewer_7jqo · 2025-08-06
> >
> > Thanks for your response and my concern is well addressed. I would raise my rating.

---

> > > ### Author Response · Authors · 2025-08-07
> > >
> > > Thank you for your insightful feedback and for taking the time to carefully evaluate our work. We truly appreciate your constructive comments on hyperparameter tuning and performance metrics. Your suggestions directly helped us strengthen the paper with more practical guidelines and detailed timing results. We’re grateful for your openness to our responses and for raising your score. Your support means a lot.

---

### Official Review · Reviewer_UTLG · 2025-07-05

**Clarity:** 3
**Significance:** 3
**Originality:** 2
**Rating:** 4
**Confidence:** 3

**Summary:**

This paper proposes Low Rank Query and Key attention (LRQK), a two-stage framework for efficient long-context inference in large language models. LRQK jointly factorizes the query and key matrices into low-rank bases during the prefill stage, and then uses these to compute proxy attention scores for top-k token selection during decoding. A hierarchical GPU–CPU caching mechanism with a hit/miss policy transfers only missing key-value pairs, preserving exact attention outputs while reducing memory and data movement costs. Extensive experiments on RULER and LongBench benchmarks with LLaMA-3 and Qwen2.5 models show LRQK achieves comparable or superior accuracy to strong sparse attention baselines while substantially lowering GPU memory consumption.

**Questions:**

List in the Weakness part

**Ethical Concerns:**

["NO or VERY MINOR ethics concerns only"]

**Final Justification:**

Most of my concerns have been addressed. I would like to retain my original positive rating.

**Limitations:**

1.Although presented algorithms are clear, actual running times, cache sizes (absolute MB/GB), and detailed hardware utilization are not given beyond high-level statements about GPU/CPU. Code is not referenced as available, and implementation details are not provided.

**Quality:**

3

**Strengths And Weaknesses:**

Strength

1. The paper addresses a real and pressing bottleneck in LLM inference: the GPU memory consumption of KV caches for long contexts. Efficient memory strategies are highly relevant for widespread LLM deployment, especially on resource-constrained devices.


Weakness：

1. The empirical comparisons primarily involve sparse/dynamic attention and offloading methods, but the selection of baselines does not cover some of the most recent quantization methods (e.g., PALU [9], which is only referenced).

[1] Efficient Low Rank Attention for Long-Context Inference in Large Language Models

[2] Scalable Efficient Training of Large Language Models with Low-dimensional Projected Attention

2. While sensitivity to rank and top-k is shown (Tables 3, 4), there is limited discussion of the overhead and tuning effort required, particularly when considering different LLM architectures

3. Results are only shown for LLaMA-3-8B and Qwen2.5-7B (Tables 1-2), lacking validation on other popular architectures.

4. Please elaborate further on the differences with other methods focusing on efficient attention for long-context inference.

5. While the paper claims to balance trade-offs between memory efficiency, numerical precision, and computational latency, it lacks experimental validation of these specific trade-offs.

---

> ### Author Rebuttal · Authors · 2025-07-31
>
> We thank the reviewer for the detailed and insightful comments.
>
> > W1. The empirical comparisons primarily involve sparse/dynamic attention and offloading methods, but the selection of baselines does not cover some of the most recent quantization methods (e.g., PALU [9], which is only referenced).
>
> Thank you for this kindness suggestion. We acknowledge that direct comparison with PALU requires more careful consideration due to the differences in methodology. PALU decomposes $\mathbf{K} \in \mathbb{R}^{l \times d}$ in dimension $d$, while our method and the comparison methods reduce both dimension $d$ and $l$.
> Following the your valuable suggestion, KVQuant is applied to both the vanilla model (`meta-llama/Llama-3.1-8B-Instruct`, `Qwen/Qwen2.5-7B-Instruct`) and the proposed method with default parameters.
> The results are shown in the table:
>
> |       RULER 32K | llama +KVQuant | llama +LRQK +KVquant | Qwen +KVQuant | Qwen +LRQK +KVQuant |
> | --------------: | -------------: | -------------------: | ------------: | ------------------: |
> |             fwe |          89.58 |                85.00 |         94.00 |               90.33 |
> |   niah_single_1 |         100.00 |               100.00 |        100.00 |              100.00 |
> |   niah_single_2 |         100.00 |               100.00 |        100.00 |               99.00 |
> | niah_multikey_1 |         100.00 |               100.00 |         96.00 |              100.00 |
> | niah_multikey_2 |          95.00 |                96.00 |         76.00 |               56.00 |
> | niah_multivalue |          95.31 |                96.75 |         70.50 |               93.50 |
> | niah_multiquery |          89.00 |                89.00 |         97.00 |               99.25 |
> |        qa_squad |          83.00 |                82.00 |         73.00 |               74.00 |
> |     qa_hotpotqa |          56.25 |                51.00 |         52.00 |               53.00 |
> |              vt |          87.50 |                95.40 |         95.40 |               90.60 |
>
> The suggested reference paper [2] is a great paper on applying low rank features on LLM training, which inspires us to explore the training side of the LLM.
> We will consider this in the future work. Thank you for this suggestion.
>
> > W2. While sensitivity to rank and top-k is shown (Tables 3, 4), there is limited discussion of the **overhead** and **tuning effort** required, particularly when considering different LLM architectures
>
> We appreciate this concern and provide comprehensive overhead analysis:
>
> **Overhead and Tuning Efforts**: Different model and dataset should have their own optimal hyperparameters.
> However, based on our experiments, there is an default choisece: $r=32$, top-$k$ = 2048, and 64 lite tokens.
> We can further tune the hyperparameters based on this default choice.
>
> > W3. Results are only shown for LLaMA-3-8B and Qwen2.5-7B (Tables 1-2), lacking validation on other popular architectures.
>
> Thank you for this valuable feedback. We have extended our evaluation to include two additional popular models: `mistralai/Mistral-7B-Instruct-v0.3` and `microsoft/Phi-3-mini-128k-instruct`. For these experiments, we used the default hyperparameters of LRQK.
> Due to memory constraints on the A6000 (48GB), the Phi-3-mini model is evaluated in 16K.
> The results are summarized in below.
>
> |       RULER 32K | Mistral | Mistral +LRQK |       RULER 16K |  Phi-3 | Phi-3 +LRQK |
> | --------------: | ------: | ------------: | --------------: | -----: | ----------: |
> |             fwe |   67.00 |         93.00 |             fwe |  92.00 |       91.00 |
> |   niah_single_1 |   98.00 |         97.00 |   niah_single_1 | 100.00 |      100.00 |
> |   niah_single_2 |   91.00 |        100.00 |   niah_single_2 | 100.00 |      100.00 |
> | niah_multikey_1 |   83.00 |         97.00 | niah_multikey_1 |  96.00 |       96.00 |
> | niah_multikey_2 |   63.00 |         51.00 | niah_multikey_2 | 100.00 |      100.00 |
> | niah_multivalue |   85.75 |         95.00 | niah_multivalue |  90.00 |       90.25 |
> | niah_multiquery |   83.75 |         93.00 | niah_multiquery |  90.50 |       87.50 |
> |        qa_squad |   59.00 |         63.00 |        qa_squad |  80.00 |       81.00 |
> |     qa_hotpotqa |   45.00 |         47.00 |     qa_hotpotqa |  51.00 |       51.00 |
> |              vt |   98.40 |         96.40 |              vt |  99.60 |       99.60 |
>
>
> > W4. Please **elaborate** further on the differences with other methods focusing on efficient attention for long-context inference.
>
> Thank you for this important suggestions.
> A comparison table highlighting key methodological differences is provided below:
>
> |           Methods |           Loki            |                                  InfiniGen                                   |                   ShadowKV                   |                   Proposed                   |
> | ----------------: | :-----------------------: | :--------------------------------------------------------------------------: | :------------------------------------------: | :------------------------------------------: |
> |         Decompose |     K(0:s) <- [K P](0:s)      |                               _, _,A = SVD(Q)                                |                A B = SVD(K)                 |      $Q K^{T} \approx A_{Q} A_{K}^{T}$       |
> |          Device K |       GPU(low rank)       |                                     GPU                                      |                  GPU(A, B)                   |             CPU(all) + GPU(hit)              |
> |          Device V |       GPU(low rank)       |                                     GPU                                      |                   CPU(all)                   |             CPU(all) + GPU(hit)              |
> | Selection Prefill | top-k of $(qP) (K P)^{T}$ | $W'\_Q, W'\_{K}$ <- top-k columns in $\lvert W\_Q \rvert, \lvert W\_K\rvert$ |                 top-k chunks                 |           top-k of $q' K'^{T}; k'$           |
> |  Selection Decode | top-k of $(q P) (KP)^{T}$ |              attention score of $q' K'^{T}$ > (max - $\alpha$)               |        top-k chunks for each KV head         |          top-k of $q' [K'; k']^{T}$          |
> |       Attention q |           $q P$           |                                      q                                       |                      q                       |                      q                       |
> |       Attention K |     $[K P]\_{\Omega}$     |                                 $K_{\Omega}$                                 | $[ A\_{\Omega} B; K\_{decode}; K^{outlier}]$ |                $K\_{\Omega}$                 |
> |       Attention V |       $V\_{\Omega}$       |                                 $V_{\Omega}$                                 |  $[ V\_{\Omega}; V\_{decode}; V^{outlier}]$  |                $V\_{\Omega}$                 |
> |        CPU <- GPU |                           |                     selected $K\_{\Omega}, V\_{\Omega}$                      |            selected $V_{\Omega}$             | hited $K\_{\Omega, hit}$, $V\_{\Omega, hit}$ |
> |             Notes | P: PCA projection matrix  |                           modified $W\_Q$, $W\_K$                            |  Extra tokens $K^{outlier}$, $V^{outlier}$   |                                              |
>
> **Distinctions:** 1. Joint decomposition, 2. Hierarchical caching, 3. Exact attention preservation.
>
> > W5. While the paper claims to balance trade-offs between memory efficiency, numerical precision, and computational latency, it lacks experimental validation of these specific trade-offs.
>
> Thank you for the feedback.
> We evaluated the trade-offs on a text summarization task using 20 samples from LongBench (context: 32K, output: up to 128 tokens), with `meta-llama/Llama-3.1-8B-Instruct`.
> The time and resource usage are measured (Nvidia 3090 24G) and shown as below:
>
> |       Method | Time (s) | tokens/s | Average GPU Util (%) | Max GPU Util (%) | GPU Memory (GB) |  GPU power (W) |
> | -----------: | -------: | -------: | -------------------: | ---------------: | --------------: | -------------: |
> |      vanilla |    17.87 |  1775.70 |       38.61;   48.38 |     99.95; 100.0 |    16.94; 17.56 | 187.24; 233.21 |
> | r=32, k=2048 |    64.84 |   489.39 |                63.64 |            100.0 |           19.82 |         203.98 |
> | r=16, k=2048 |    57.04 |   556.22 |                65.29 |            100.0 |           18.85 |         211.96 |
> | r=32, k=1024 |    52.62 |   603.05 |                56.86 |            100.0 |           19.51 |         199.59 |
> | r=16, k=1024 |    57.04 |   646.33 |                51.15 |            100.0 |           19.33 |         199.85 |
>
>
> > L1. Although presented algorithms are clear, actual running times, cache sizes (absolute MB/GB), and detailed hardware utilization are not given beyond high-level statements about GPU/CPU. Code is not referenced as available, and implementation details are not provided.
>
> We acknowledge this limitation and will provide detailed implementation metrics:
>
> - **Cache in GPU**: 1. The most recently accessed KV cache, 2. Low rank matrices $\mathbf{A}\_{K}, \mathbf{B}\_{K}, \mathbf{B}\_{Q}$, 3. The weights of LLM.
> - **Cache in CPU**: 1. The full KV cache.
> - **Actual running times**: Please refer to response W5.
> - **Implementation**:  A lightweight version of our code is provided in the supplementary material. Key aspects include:
>   - Built on the standard Hugging Face `transformers` library.
>   - The original attention mechanism is replaced with our proxy attention mechanism.
>   - A self-defined buff session is introduced to manage the cache.
>   - Two core algorithms are further compiled with `torch.compile`.

---

> > ### Comment · Reviewer_UTLG · 2025-08-02
> >
> > Thank you for your response and for addressing my questions. After reading the authors’ response, many of my earlier concerns have been resolved. Here is my updated assessment of the concerns I raised earlier:
> >
> > Q1: The baseline comparison lacks recent quantization methods such as PALU.
> >
> > Now: Almost resolved. The authors not only acknowledged the issue but provided new experiments.
> >
> > Q2: Limited discussion of overhead and tuning costs for different architectures.
> >
> > Now: Not a concern. The authors provided practical defaults.
> >
> > Q3: Evaluation is limited to LLaMA3-8B and Qwen2.5-7B.
> >
> > Now: Not a concern.
> >
> > Q4: Please elaborate on differences from other efficient attention methods.
> >
> > Now: Not a concern.
> >
> > Q5: Claims about trade-offs are not empirically supported.
> >
> > Now: Resolved. Authors provided quantitative data covering all trade-off dimensions, strengthening practical claims.
> >
> > L1: No concrete runtime/cache/hardware info or code reference.
> >
> > Now: Implementation info is reasonably clear.

---

### Official Review · Reviewer_GeRa · 2025-07-06

**Clarity:** 3
**Significance:** 3
**Originality:** 3
**Rating:** 5
**Confidence:** 4

**Summary:**

The paper proposes Low-Rank Query and Key attention (LRQK), a two-stage framework for reducing the KV-cache memory footprint during long-context inference in decoder-only large language models (LLMs). For a long prompt, the full-precision query (Q) and key (K) matrices are jointly factorised into rank-r bases by minimising a reconstruction loss (Eq. 3). At each generation step, low-rank projections estimate “proxy” attention scores in O(ℓr) time; only the top-k tokens plus a small recency buffer are kept on-GPU, while the rest of the cache resides on CPU. A hit/miss mechanism fetches full-precision KV rows for missed indices, guaranteeing exact attention on the selected subset. Experiments with LLaMA-3-8B-1M and Qwen-2.5-7B on RULER-128 K and LongBench show that LRQK matches or beats shadow-KV, InfiniGen, QUEST and Loki on most tasks, e.g., +2 pp on QA-SQuAD versus the 8B baseline while cutting CPU↔GPU transfers by ≈60 % on WikiText-2 . Hyper-parameter sweeps reveal favourable rank/top-k trade-offs (r = 32 and k = 1024 give near-lossless accuracy)

**Questions:**

1. Section 5 reports “≈ 60 % reduction in CPU↔GPU transfers” but gives no tokens-per-second or latency figures on the A100 node you mention (lines 210-216). Could you provide end-to-end throughput (tokens/s) and peak GPU memory for LRQK vs. ShadowKV, InfiniGen and vanilla attention at 128 K tokens?

2. All experiments use 7-8 B parameters. Are there bandwidth or PCIe limits that prevent LRQK from scaling to 30 B+ or mixture-of-experts LLMs where KV memory dominates? Even a profiling statement would help assess significance.

3. Since LRQK already uses low-rank factors, can the retained full-precision KV rows be post-quantised to 8-bit or 4-bit (à la KVQuant) without harming accuracy? Discussing complementarity with quantisation would broaden impact.

**Ethical Concerns:**

["NO or VERY MINOR ethics concerns only"]

**Final Justification:**

The authors have sufficiently addressed all the concerns I have raised during the rebuttal. I think this paper tackles a timely challenge of solving the memory constraints in long context inference. The novel techniques and the findings presented in the paper will be valuable for the entire community.

**Limitations:**

The authors candidly notes that rank r and active-set size k require task-specific tuning and that optimal values differ across architectures. However, the societal-impact section is missing. Please add a brief discussion on potential misuse (e.g., enabling cheaper deployment of disinformation bots) and mitigation strategies.

**Quality:**

3

**Strengths And Weaknesses:**

Strengths:
1. This paper provides a novel and innovate approach to solving the memory constraints in long context inference. Unlike many eviction or sparsity methods, the hit/miss fetch ensures identical outputs to full attention on the retained tokens—no approximation error.
2. The paper is generally well written and each technique introduced comes with proper motivation. This paper is timely and looks at some key memory issues with kv-caching.
3. Authors provide strong results. With LLaMA-3-8B-1M at 128 K tokens, LRQK matches or beats strong baselines (ShadowKV, Quest, InfiniGen, Loki) on many RULER tasks and improves Passage-Retrieval and LCC on LongBench (Table 1). I also found the memory-transfer savings results interesting like on WikiText-2 the hit/miss cache cuts CPU ↔ GPU KV transfers by ≈60 % across a broad grid of settings (mean miss-rate ≈0.40, Fig. 5) Moreover, authors have done extensive ablations on rank r, active-set size k, miss-rate, and cache policy across two architectures and multiple long-context benchmarks.
4. Authors promised to open source the code which will be valuable to the community.

Weaknesses:
1. Experiments list the hardware (A100) but omit tokens/s throughput, wall-clock latency, and peak/resident GPU memory, making it hard to judge real deployment benefits
2. All results use LLaMA-3-8B-1M and Qwen-2.5-7B; the method’s scalability to 30 B + or MoE LLMs, where KV caches are a bigger bottleneck, remains unverified. The authors themselves flag that rank r and active-set size k require dataset/model-specific tuning, and optimal values can vary across tasks. I am concerned about the ease of adoption of this technique for more general use cases.
3. Equation 3 formulation is internally inconsistent. The hard constraints $ Q=A_QB_Q, K=A_KB_K $ in Eq. 3 cannot hold for low rank $r<d$ and are immediately relaxed in the Lagrangian; also, the interaction term omits $B_QB_K^T$. This is mostly cosmetic but weakens the mathematical presentation.

---

> ### Author Rebuttal · Authors · 2025-07-31
>
> Thank the reviewer for insightful comments.
>
> > W1. Experiments list the hardware (A100) but omit tokens/s throughput, wall-clock latency, and peak/resident GPU memory, making it hard to judge real deployment benefits
>
> Thank you for this valuable feedback. We provide comprehensive timing results on single NVIDIA A100 40GB with Llama-3-8B-1M.
> The following table shows the results. Further details can be found in the supplemental material.
>
> |           Method |       4K |    4K |       8K |    8K |      16K |   16K |      32K |   32K |     64K |  64K |
> | ---------------: | -------: | ----: | -------: | ----: | -------: | ----: | -------: | ----: | ------: | ---: |
> |                  |     TTFT |  TPOT |     TTFT |  TPOT |     TTFT |  TPOT |     TTFT |  TPOT |    TTFT | TPOT |
> |         GPU only | 37500.29 | 35.40 | 37734.19 | 35.64 | 33649.79 | 35.36 | 32111.45 | 35.77 |     OOM |  OOM |
> |      CPU offload |  6945.10 |  4.31 |  6984.35 |  4.32 |  7073.53 |  2.40 |  4849.19 |  0.98 | 4131.00 | 0.50 |
> |     LRQK default |  7120.49 |  5.68 |  7280.39 |  5.72 |  6379.91 |  5.49 |  5436.66 |  5.36 | 4103.05 | 6.80 |
> | LRQK no hit/miss |  7180.64 |  2.50 |  6747.19 |  2.53 |  5317.06 |  2.31 |  4920.02 |  2.48 | 4163.86 | 1.95 |
>
>
> > W2. All results use LLaMA-3-8B-1M and Qwen-2.5-7B; the method’s scalability to 30 B + or MoE LLMs, where KV caches are a bigger bottleneck, remains unverified. The authors themselves flag that rank r and active-set size k require dataset/model-specific tuning, and optimal values can vary across tasks. I am concerned about the ease of adoption of this technique for more general use cases.
>
> Thank you for this insightful question. We provide further validation on larger models: `Qwen/Qwen2.5-14B-Instruct` (64K context) and `Qwen/Qwen2.5-32B-Instruct` (16K context).
> YaRN is applied on 64K context. LRQK use default settings.
> Results are shown below.
>
> |       RULER 64K | Qwen 14B | Qwen 14B +LRQK |       RULER 16K | Qwen 32b | Qwen 32B +LRQK |
> | --------------: | -------: | -------------: | --------------: | -------: | -------------: |
> |             fwe |    90.33 |          78.33 |             fwe |    94.67 |          94.00 |
> |   niah_single_1 |    57.00 |          99.00 |   niah_single_1 |   100.00 |         100.00 |
> |   niah_single_2 |    55.00 |          96.00 |   niah_single_2 |   100.00 |         100.00 |
> | niah_multikey_1 |    46.00 |          80.00 | niah_multikey_1 |   100.00 |         100.00 |
> | niah_multikey_2 |    24.00 |          22.00 | niah_multikey_2 |    99.00 |          99.00 |
> | niah_multivalue |    54.00 |          82.25 | niah_multivalue |   100.00 |         100.00 |
> | niah_multiquery |    49.75 |          88.50 | niah_multiquery |   100.00 |         100.00 |
> |        qa_squad |    64.00 |          50.00 |        qa_squad |    86.00 |          86.00 |
> |     qa_hotpotqa |    41.00 |          36.00 |     qa_hotpotqa |    66.00 |          66.00 |
> |              vt |    57.80 |          97.20 |              vt |    86.40 |          82.60 |
>
> The table shows that the proposed method can be applied to larger models.
>
> > W3. Equation 3 formulation is internally inconsistent. The hard constraints $Q=A\_Q B\_Q, K = A\_K B\_K$ in Eq. 3 cannot hold for low rank $r<d$ and are immediately relaxed in the Lagrangian; also, the interaction term omits $B\_Q B\_K^T$. This is mostly cosmetic but weakens the mathematical presentation.
>
> We acknowledge the reviewer's concern and provide the following clarification:
> You are correct that the hard constraints $Q = A\_Q B\_Q$ and $K = A\_K B\_K$ cannot hold exactly when $r < d$. The formulation should indeed reflect that these are approximation constraints. We will revise the formulation to be more mathematically precise:
>
> $$
> \mathop{argmin}_{A\_{Q}, B\_{Q}, A\_{K}, B\_{K}} \dfrac{1}{2} ||
>   Q K^T - A\_{Q} A\_{K}^T
>   ||_F^2,
>   \text{where }
>   Q \approx A\_{Q} B\_{Q},
>   K \approx A\_{K} B\_{K}.
> $$
>
> Regarding the interaction term omitting $B\_Q B\_K^T$: The key insight is that we are **not** approximating $Q K^T$ as $(A\_Q B\_Q)(A\_K B\_K)^T = A\_Q B\_Q B\_K^T A\_K^T$.
>
> Instead, our approach recognizes that:
> 1. Both $Q$ and $K$ have low-rank structure individually
> 2. Their product $Q K^T$ can be approximated more efficiently by directly learning compact representations $A\_Q$ and $A\_K$ such that $Q K\^T \approx A\_Q A\_K^T$
>
> We will revise the manuscript to clarify these mathematical details and ensure the formulation is both precise and well-motivated.
>
> > Q1. Section 5 reports “≈ 60 % reduction in CPU <-> GPU transfers” but gives no tokens-per-second or latency figures on the A100 node you mention (lines 210-216). Could you provide end-to-end throughput (tokens/s) and peak GPU memory for LRQK vs. ShadowKV, InfiniGen and vanilla attention at 128 K tokens?
>
> Thank you for this important question.
> We provide a timing results on NVIDIA 3090 24GB under text summarization task (20 samples from LongBench, context 32K, output up to 128 tokens) with `meta-llama/Llama-3.1-8B-Instruct` as the base model.
> Due to time limit, we evaluate the vanilla attention (2 GPUs) and ShadowKV (1 GPU), and the proposed method (1 GPU).
> Average results are shown below:
>
> |             Method | Time (s) | tokens/s | Average GPU Util (%) | Max GPU Util (%) | GPU Memory (GB) |  GPU power (W) |
> | -----------------: | -------: | -------: | -------------------: | ---------------: | --------------: | -------------: |
> |            vanilla |    17.87 |  1775.70 |       38.61;   48.38 |     99.95; 100.0 |    16.94; 17.56 | 187.24; 233.21 |
> |           shadowkv |    29.29 |  1083.55 |                58.00 |            100.0 |           17.93 |         243.85 |
> | LRQK(r=32, k=2048) |    64.84 |   489.39 |                63.64 |            100.0 |           19.82 |         203.98 |
> | LRQK(r=16, k=2048) |    57.04 |   556.22 |                65.29 |            100.0 |           18.85 |         211.96 |
> | LRQK(r=32, k=1024) |    52.62 |   603.05 |                56.86 |            100.0 |           19.51 |         199.59 |
> | LRQK(r=16, k=1024) |    57.04 |   646.33 |                51.15 |            100.0 |           19.33 |         199.85 |
>
>
> > Q2. All experiments use 7-8 B parameters. Are there bandwidth or PCIe limits that prevent LRQK from scaling to 30 B+ or mixture-of-experts LLMs where KV memory dominates? Even a profiling statement would help assess significance.
>
> **Bottle neck**:
> PCIe limits and GPU bandwidth is not the main limitation, since the data need to be transfered between CPU and GPU is relatively small.
> Based on our observation, the main time consumption is the indexing operation in CPU. It takes time to fetch and collect hitted tokens in CPU. We are looking for a better method to reduce this time consumption.
>
> > Q3. Since LRQK already uses low-rank factors, can the retained full-precision KV rows be post-quantised to 8-bit or 4-bit (à la KVQuant) without harming accuracy? Discussing complementarity with quantisation would broaden impact.
>
> Thank you so much for this sugessions, the quantization methods are valuable to explore.
> Following the your suggestion, KVQuant is evaluated to both the vanilla model (`meta-llama/Llama-3.1-8B-Instruct` and `Qwen/Qwen2.5-7B-Instruct`) and the model with proposed method. The results are shown in the following table.
> The hypterparameters of LRQK are $r=32$, top-$k$ = 2048, and 64 lite tokens.
>
> |       RULER 32K | llama +KVQuant | llama +LRQK +KVquant | Qwen +KVQuant | Qwen +LRQK +KVQuant |
> | --------------: | -------------: | -------------------: | ------------: | ------------------: |
> |             fwe |          89.58 |                85.00 |         94.00 |               90.33 |
> |   niah_single_1 |         100.00 |               100.00 |        100.00 |              100.00 |
> |   niah_single_2 |         100.00 |               100.00 |        100.00 |               99.00 |
> | niah_multikey_1 |         100.00 |               100.00 |         96.00 |              100.00 |
> | niah_multikey_2 |          95.00 |                96.00 |         76.00 |               56.00 |
> | niah_multivalue |          95.31 |                96.75 |         70.50 |               93.50 |
> | niah_multiquery |          89.00 |                89.00 |         97.00 |               99.25 |
> |        qa_squad |          83.00 |                82.00 |         73.00 |               74.00 |
> |     qa_hotpotqa |          56.25 |                51.00 |         52.00 |               53.00 |
> |              vt |          87.50 |                95.40 |         95.40 |               90.60 |
>
> From this table, we can find that in general applying KVQuant with the proposed method does not siginificantly harm the performance.
>
> > L1. The authors candidly notes that rank r and active-set size k require task-specific tuning and that optimal values differ across architectures. However, the societal-impact section is missing. Please add a brief discussion on potential misuse (e.g., enabling cheaper deployment of disinformation bots) and mitigation strategies.
>
> Thank you so much for this valuable sugessions.
>
> **Hyperparameter Tuning**: Different models and datasets may require unique set of optimal hyperparameters.
> However, there is a default choice that can help reduce the tuning effort.
> Based on our experiments, we can further turn from a default choice: $r=32$, top-$k$ = 2048, and 64 lite tokens.
> For instance, the rank $r$ can be chosen from $\{8, 16, 32, 48\}$, and the top-$k$ value can be adjusted to accommodate different context lengths.
>
> **Societal impact**:
> LRQK's reduced inference costs could enable broader access to long-context language models, benefiting research and education. However, this efficiency may also lower barriers to deploying harmful applications such as large-scale disinformation campaigns.
> We believe democratized access to advanced AI capabilities provides net benefits when coupled with appropriate safeguards and responsible deployment practices.

---

> > ### Author Response · Authors · 2025-08-05
> >
> > Dear Reviewer GeRa,
> >
> > Thank you for your valuable comments.
> >
> > We have carefully addressed the concerns raised in your review. A summary of our responses is as follows:
> > - **System Performance Metrics**: The results on the tokens/s throughput, latency, and memory usage have been added (See reply to W1 and Q1).
> > - **Experiments on Larger Models**: The experiments on larger models (14B and 32B) have been added (See reply to W2).
> > - **Mathematical Presentation**: The mathematical presentation has been improved (See reply to W3).
> > - **Quantization Methods**: The quantization methods (KVQuant) have been evaluated (See reply to Q3).
> > - **Hyperparameter Tuning and Societal impact**: The hyperparameter tuning and societal impact have been addressed (See reply to L1).
> >
> > If you have any further questions or concerns, we would be happy to discuss them. We sincerely appreciate the time and effort you have devoted to reviewing our work.

---

### Note · Authors · 2025-08-12

Dear Reviewers and AC,

We sincerely thank you for your thoughtful feedback, constructive suggestions, and continued support throughout the review process. Your insights have significantly strengthened the rigor, clarity, and practical value of the work.

Key points addressed in the rebuttal are briefly summarized below:
1. **Hyperparameter Tuning** (Reviewers GeRa, UTLG, 7jqo, 43cG): Detailed guidelines for hyperparameter selection and a set of recommended default values are provided, enhancing reproducibility and ease of adoption.
2. **System Performance Metrics** (Reviewers GeRa, UTLG, 7jqo): Critical system-level measurements are included, such as latency, throughput, and memory usage, that are essential for real-world deployment.
3. **Implementation Details and Overheads** (Reviewers UTLG, 43cG): The implementation strategies and overheads are carefully discussed.
4. **Validation Across LLMs** (Reviewers GeRa, UTLG): The evaluation on additional models are included: `mistralai/Mistral-7B-Instruct-v0.3`, `microsoft/Phi-3-mini-128k-instruct`, `Qwen/Qwen2.5-14B-Instruct`, and `Qwen/Qwen2.5-32B-Instruct`, confirming the generality of the proposed method.
5. **Evaluation with Quantization** (Reviewers GeRa, UTLG): New results with KVQuant are also included, demonstrating effectiveness under quantized settings.
6. **Other Comments**: Additional clarifications and responses to other comments are provided to ensure comprehensive understanding, for example, the mathematical representation, the comparison with other methods, etc.

We appreciate the time and care you have invested in reviewing the paper. We will carefully addressed all concerns and will incorporate the feedback into the final version. Thank you again for your valuable guidance.

---

### Decision · Program_Chairs · 2025-09-17

**Decision:**

Accept (poster)

**Comment:**

LRQK is a novel approach with exact attention preservation. It shows strong experimental results, outperforming sparse-attention methods and achieving significant memory savings.

Initial paper lacked performance metrics and scalability validation for larger models. Hyperparameter guidelines were insufficient and implementation overheads unclear.

The paper offers an Innovative solution to a critical LLM memory bottleneck. Authors' thorough rebuttal addressed all major initial weaknesses with new experiments and clarifications, strengthening the paper.

Reviewers questioned metrics, scalability, hyperparameter tuning, and implementation. Authors added detailed timing/memory results, validated on larger models, and provided hyperparameter guidelines/implementation details. All concerns resolved, leading to acceptance.